# INFORMATIVE POSTERIOR ENSEMBLES FOR SEQUENTIAL SIMULATION-BASED INFERENCE

## ABSTRACT

Approximating parameter posteriors in likelihood-free settings is a practical challenge common to many scientific disciplines. While recent advances in both computer simulation and generative modeling have paved the way for tractable inference in high-fidelity environments, they often require prohibitively large sample sizes in practice. Sequential posterior estimation methods attempt to mitigate this by iteratively producing proposal distributions that refine the inverse model, but they lack explicit selection mechanisms for reducing information overlap in proposed simulations. In this work, we introduce a mutual information-based acquisition scheme for identifying informative simulation parameters, operating on disagreement in the parameter space across a weighted posterior ensemble of atomic proposals. Our approach crucially leverages only an inverse model, making it compatible with existing direct posterior estimation procedures. We further extend this approach to the batched setting and formulate a fast approximate algorithm that preserves posterior convergence guarantees. We demonstrate the potential of this method on several common simulation-based inference (SBI) benchmark tasks, and observe performance advantages over non-ensemble counterparts in low-data regimes.

## 1   INTRODUCTION

The intractability of likelihood functions is a common barrier to Bayesian inference in complex, real-world settings. While high-fidelity simulators may be readily available as models of underlying generative processes, they often do not admit a closed-form likelihood. Simulation-based inference (SBI) methods work around this limitation by assuming such models can only *generate* samples, and attempt to learn a posterior distribution from the resulting simulation data (Cranmer et al., 2020).

Early success in this direction involved easy-to-use methods such as Approximate Bayesian Computation (ABC) and extensions of kernel density estimation (Rubin, 1984; Beaumont et al., 2002; Sisson et al., 2007; Marjoram et al., 2003). These methods struggle to scale with the dimensionality of most real-world applications, however, and neural network-based methods (Papamakarios & Murray, 2016; Lueckmann et al., 2017; Greenberg et al., 2019) have since been proposed to better address this challenge. Newer methods offer greater flexibility in the approximated probabilistic form used by the inference pipeline, namely the posterior (Papamakarios & Murray, 2016; Deistler et al., 2022), the likelihood (Papamakarios et al., 2019; Lueckmann et al., 2019), and likelihood ratio (Hermans et al., 2020; Durkan et al., 2020). These methods also support a variety of underlying neural density estimators (NDEs), including mixture density networks (Bishop, 1994) and normalizing flows Rezende & Mohamed (2016); Papamakarios et al. (2021), along with popular extensions (e.g., Real NVP Dinh et al. (2017), MAE Germain et al. (2015), MAF Papamakarios et al. (2018), etc). Diffusion-based approaches leverage score estimation and flow matching (Geffner et al., 2023a; Sharrock et al., 2022; Dax et al., 2023) to the same end, but offer different tradeoffs in scalability and efficiency.

In this work, we introduce the following:

1. An ensembling scheme for managing a collection of posterior models across several rounds of inference. This scheme provides a tractable means of updating and combining NDEs $q_\phi(\theta|x)$ into valid atomic proposals under the model weight posterior $p(\phi|D)$.

2. A mutual information-based acquisition scheme that prioritizes parameters expected to reduce NDE model uncertainty while preserving posterior convergence guarantees. This approach operates under a notion of *residual* mutual information and critically does not require a surrogate likelihood (forward) model. We provide theoretical foundations for this approach and describe how it can be implemented efficiently in practice.

3. A fully batched version of the mutual information-based scheme described above. Sequential SBI methods often collect simulation data in parallel, and therefore must commit to batches of parameter samples in between model updates. The proposed batch acquisition function provides a holistic means of addressing high information overlap over samples from the proposal distribution by modeling the joint effects of candidate batches. We further describe a greedy $(1 - 1/e - \epsilon)$-approximate random algorithm for computing the optimal batch.

We evaluate the proposed methods on benchmark datasets against state-of-the-art SBI baselines and demonstrate that our approach achieves superior performance compared to existing posterior-direct methods (e.g., SNPE), requiring only a marginal increase in computational cost. These results highlight the practicality of our method for real-world tasks where sample efficiency is critical, and we provide ablations to clarify the cost-to-performance tradeoffs for practitioners navigating the method landscape.

## 2 BACKGROUND

### 2.1 NEURAL POSTERIOR ESTIMATION

Simulation-based inference seeks to approximate the posterior distribution $p(\theta|x)$ under a stochastic model $p(x|\theta)$. We assume $p(x|\theta)$ is defined implicitly via a simulation-based program, and while samples $x \sim p(x|\theta)$ can be drawn, direct evaluation of the likelihood value $p(x|\theta)$ is not possible. We further focus on the sequential (non-amortized) case, wherein an observational data point of interest $x_o$ is known ahead of time, and we place particular emphasis on learning a high-quality approximation of $p(\theta|x_o)$.

Neural Posterior Estimation (NPE) methods approximate the posterior distribution directly by training an NDE $q_\phi(\theta|x)$ via maximum likelihood on samples $\{(\theta_i, x_i)\}_{i=1}^N$, where $\theta_i \sim p(\theta)$ and $x_i \sim p(x|\theta_i)$, i.e., by minimizing the loss

$$\mathcal{L}(\phi) = \mathbb{E}_{\theta \sim p(\theta)} \mathbb{E}_{x \sim p(x|\theta)} \left[ -\log q_\phi(\theta|x) \right], \tag{1}$$

where $\phi$ are the model's learnable parameters. So long as $q_\phi$ is sufficiently expressive, by Proposition 1 of Papamakarios & Murray (2016) $q_\phi(\theta|x)$ will converge to the true posterior $p(\theta|x)$ in the limit as $N \to \infty$.

Sequential Neural Posterior Estimation (SNPE) methods divide the inference process into multiple iterations of NPE, improving sample efficiency by leveraging the observation that $p(\theta|x = x_o)$ is typically much narrower than $p(\theta)$. SNPE methods leverage this by drawing $\theta$ values expected to be more informative about $p(\theta|x_o)$, using a proposal distribution $\tilde{p}(\theta)$ at each round that reflects the model's current posterior approximation $q_\phi(\theta|x = x_o)$. Training an NDE $q_\phi$ on samples $\theta \sim \tilde{p}(\theta)$ when $\tilde{p}$ differs from the true prior results in convergence to a proposal posterior,

$$\tilde{p}(\theta|x) = p(\theta|x) \frac{\tilde{p}(\theta)p(x)}{\tilde{p}(x)p(\theta)}, \tag{2}$$

rather than the true posterior (where $\tilde{p}(x) = \int_\Theta p(x|\theta)\tilde{p}(\theta)d\theta$). Different SNPE variants correct for this in distinct ways: *SNPE-A* (Papamakarios & Murray, 2016) trains $q_\phi(\theta|x)$ to approximate $\tilde{p}(\theta|x)$ at each round and applies importance reweighting afterward; *SNPE-B* (Lueckmann et al., 2017) minimizes an importance-weighted loss directly alongside calibration kernels and Bayesian mixture density networks; and *SNPE-C* (Greenberg et al., 2019), also known as Automatic Posterior Transformation (APT), enables directly training posterior models under flexible atomic proposals.

### 2.2 ATOMIC PROPOSALS WITH APT

Greenberg et al. (2019) observe that minimizing the loss $\tilde{\mathcal{L}}(\phi) = -\sum_{i=1}^N \log \tilde{q}_\phi(\theta_i|x)$, where

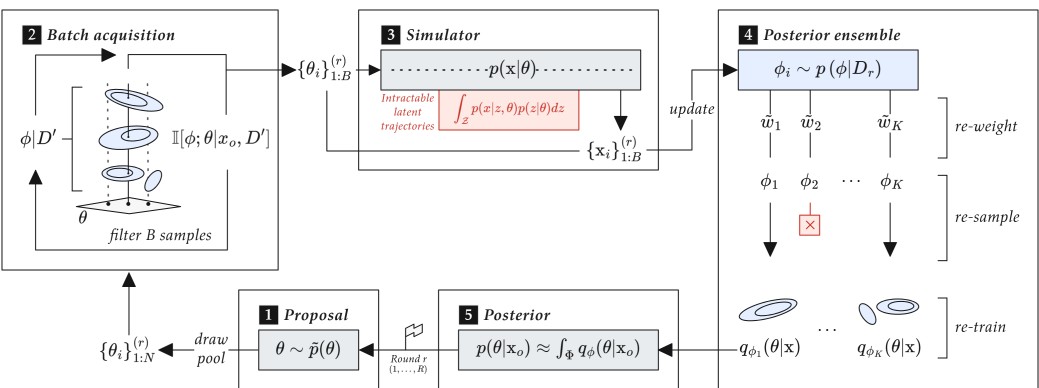

Figure 1: **Depiction of the batch active ESNPE pipeline.** Candidate parameters $\theta_i$ are drawn the current round's proposal $\tilde{p}(\theta)$, filtered to a sample of size $B$ according to Alg. 2, and run through the simulator $p(x|\theta)$ to generate pairs $\{(\theta_i, x_i)\}_{1:B}^{(r)}$ in round $r$. The posterior ensemble is subsequently reweighted, resampled, and retrained (see Appendix E). The next round's proposal $\tilde{p}_{r+1}(\theta)$ is then produced by conditioning the ensembled posterior on the target observation $x_o$.

$$\tilde{q}_\phi(\theta|x) = \frac{q_\phi(\theta|x)\,(\tilde{p}(\theta)/p(\theta))}{\int_\Theta q_\phi(\theta|x)\,(\tilde{p}(\theta)/p(\theta))\,d\theta} \tag{3}$$

produces $q_\phi(\theta|x) \to p(\theta|x)$ as $N \to \infty$. This is (again) by virtue of Prop. 1 of Papamakarios & Murray (2016); $\tilde{\mathcal{L}}(\phi)$ is minimized only when $\tilde{q}_\phi(\theta|x) = \tilde{p}(\theta|x)$, and thus $q_\phi(\theta|x) = p(\theta|x)$ by Eq. 2. The authors further extend this scheme to arbitrary "atomic proposals," subverting the need for closed-form normalization constants in $\tilde{q}_\phi(\theta|x)$. The atomic loss scheme sets a uniform proposal prior $\tilde{p}(\theta) = U_\Theta$ over a fixed batch of parameters $\Theta = \{\theta_1, \ldots, \theta_M\}$, producing a categorical $\tilde{q}_\phi(\theta|x)$:

$$\tilde{q}_\phi(\theta|x) = \frac{q_\phi(\theta|x)/p(\theta)}{\sum_{\theta' \in \Theta} q_\phi(\theta'|x)/p(\theta')} \tag{4}$$

The loss $\tilde{\mathcal{L}}$ now no longer relies on any earlier choice of proposal prior (once $\Theta$ has been fixed), and as before, $\mathbb{E}_{\theta \sim U_\Theta, x \sim p(x|\theta)}[\tilde{\mathcal{L}}]$, is minimized when $\tilde{q}_\phi(\theta|x)$ is the true proposal posterior $\tilde{p}(\theta|x)$.

## 3 METHODOLOGY

While APT ensures the NDE model converges to the true posterior in the limit as $N \to \infty$, in practice we often face several challenges: the NDE can get "stuck" in certain regions of the parameter space or require a large number of samples before the true posterior shape begins to emerge. We first seek to stabilize round-by-round posterior estimates by ensembling several posterior approximations, and do so in a manner that preserves posterior convergence while maintaining component alignment with the model weight posterior $p(\phi|D)$.

### 3.1 SEQUENTIAL POSTERIOR ENSEMBLES

Given access to the posterior $p(\phi|D)$ over NDE models $q_\phi(\theta|x)$, one can marginalize over $\phi$ to produce a posterior predictive distribution

$$q_\Phi(\theta|x, D) = \int_\Phi q_\phi(\theta|x) p(\phi|D) d\phi, \tag{5}$$

independent of any particular selection of model weights $\phi$, conditional on all observed simulation data $D$. Access to $p(\phi|D)$ further enables uncertainty quantification over the learned models, e.g., variance in the posterior estimate $\mathbb{V}_{\phi|D}[q_\phi(\theta|x)]$ as explored in Järvenpää et al. (2019); Lueckmann et al. (2019); Griesemer et al. (2024); Krouglova et al. (2025), but exhibit limitations that prevent existing methods from applying to atomic proposals in sequential inference settings (detailed discussion

---

**Algorithm 1** Ensemble Sequential Neural Posterior Estimation (ESNPE)

---

**Input:** Prior $p(\theta)$, target observation $x_o$, round-wise selection size $B$, round-wise sample size $N$, total rounds $R$, inference prior $p(\theta)$, model prior $p(\phi)$
**Output:** Ensemble posterior approximation $\bar{q}_\Phi(\theta|\mathrm{x}_o)$

    Let $D^{(0)} = \{\}$
    Let $\tilde{p}^{(0)}(\theta) = p(\theta)$
    Initialize $K$ NDEs $\phi_1, \ldots, \phi_K \sim p(\phi)$
    **for** $r \in [1, \ldots, R]$ **do**
        Draw $N$ samples $\{\theta_i\}_{1:N} \sim \tilde{p}^{(r-1)}(\theta)$
        Actively acquire parameter batch $\{\theta_i\}_{1:B}$         // (optional) see Alg. 2
        **for** $b \in [1, \ldots, B]$ **do**
            Simulate $x_b \sim p(x|\theta_b)$
            Set $D^{(r-1)} = D^{(r-1)} \cup \{(\theta_b, x_b)\}$
        Set $D^{(r)} = D^{(r-1)}$
        **for** $k \in [1, \ldots, K]$ **do**
            Compute SNIS weight $\tilde{w}_k^{(r)}$         // see Eq. 7
            Resample and rejuvenate $q_{\phi_k}$         // see Appendix E
        Let $\bar{q}_\Phi^{(r)}(\theta|x) = \sum_{j=1}^K \tilde{w}_j^{(r)} q_{\phi_j}(\theta|x)$
        Set $\tilde{p}^{(r)}(\theta) = \bar{q}_\Phi^{(r)}(\theta|x_o)$
    **return** $\bar{q}_\Phi^{(R)}(\theta|\mathrm{x}_o)$

---

provided in Appendix F.2). We therefore seek a flexible means of approximating $p(\phi|D)$ via an ensemble of atomic proposal distributions $q_{\phi_i}(\theta|x)$ which can be updated under the most recent observations, i.e., $p(\phi|D^{(r)})$ at each round $r$.

Notice that, via Bayes' rule, for a set of observations $\{(\theta_i', x_i')\}_{i=1}^N$,

$$p(\phi|D') = \frac{p(\phi|D) \prod_i^N p(\theta_i'|x_i', \phi)}{\int_\phi p(\phi|D) \prod_i^N p(\theta_i'|x_i', \phi) d\phi}, \tag{6}$$

where $D' = D \cup \{(\theta_i', x_i')\}_{i=1}^N$. A model $\phi \sim p(\phi|D)$ can therefore be brought up-to-date, in terms of its likelihood $p(\phi|D')$, by calculating its (normalized) support under the new pairs $D' \setminus D$. This update can be approximated under a collection of $K$ models with self-normalized importance weights (SNIS; Elvira & Martino (2021)); we want our models to appear as if drawn from $p(\phi|D')$ when they are in fact from $p(\phi|D)$. For the $j$th component model $q_j(\theta|x)$, we calculate its updated weight under new simulation data $D^{(r)} = D \cup \{(\theta_i^{(r)}, x_i^{(r)})\}_{i=1}^{N_r}$ observed at round $r$ as

$$\tilde{w}_j^{(r)} = \frac{p(\phi_j|D) \prod_i^{N_r} q_{\phi_j}\left(\theta_i^{(r)}|x_i^{(r)}\right)}{\sum_{k=1}^K p(\phi_k|D) \prod_i^{N_r} q_{\phi_k}\left(\theta_i^{(r)}|x_i^{(r)}\right)}, \tag{7}$$

where $\tilde{w}_j^{(r)} \approx p(\phi_j|D^{(r)})$, and is asymptotically consistent as $K \to \infty$. The ensemble posterior model for round $r$ is then $\bar{q}_\Phi^{(r)}(\theta|x) = \sum_{j=1}^K \tilde{w}_j^{(r)} q_{\phi_j}(\theta|x)$, which serves as an approximation to the posterior predictive $q_\Phi(\theta|x, D)$. Each component model can individually be retrained under the APT loss from Eq. 4 over the shared, fixed batch $\Theta^{(r+1)} \sim \tilde{p}^{(r+1)}$ (see Appendix D.1 for more on atomic ensembles). We further set the next round's proposal $\tilde{p}^{(r+1)}(\theta) = \bar{q}_\Phi(\theta|x_o, D)$ and repeat until the simluation budget is reached. This routine is summarized in Algorithm 1.

Note that computing weight updates for large batches can be challenging in practice, often resulting in a low effective sample size (ESS) and assigning all weight to a single model. Further, model diversity is key after reweighting, and retraining under the standard atomic loss can hinder the posterior coverage of $p(\phi|D)$. We introduce an augmented atomic loss under weighted likelihood bootstraps (WLB) (Newton & Raftery, 1994) to better rejuvenate ensemble components after each round and combat premature homogeneity. See Appendix E additional details.

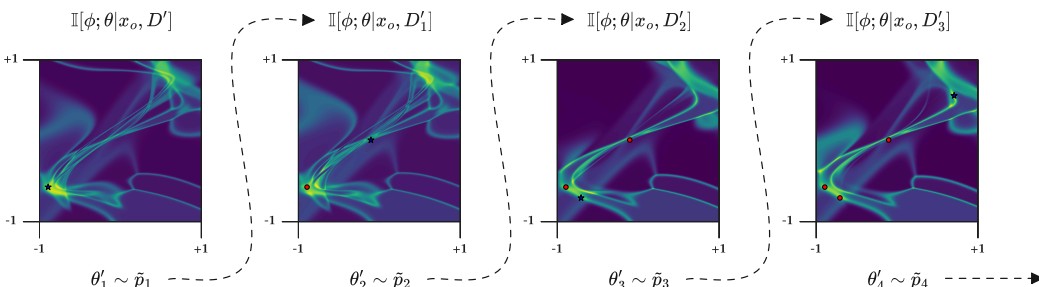

Figure 2: **Iterative batch-wise parameter selection.** This diagram depicts the mutual information-adjusted proposal $\tilde{p}_i$ as it is actively updated during the batch collection process. The proposals shown are taken from an intermediate round of a run reported in Table 1 for *Batch MI ESNPE* on the *Two Moons* task, and the plots show proposal likelihoods over the prior support. Blue stars indicate the parameter $\theta_i \sim \tilde{p}_i$ selected at the $i$-th stage in the batch, while red points show previously collected points.

---

**Algorithm 2** Random $(1 - 1/e - \epsilon)$-approximate joint MI batch collection algorithm

---

**Input:** Unweighted sample $\phi_1, \ldots, \phi_K \sim p(\phi|D)$, batch size $B$, candidate parameters $\{\theta_1, \ldots, \theta_N\}$, dataset $D$
**Output:** Parameter batch $S = \{\theta'_1, \ldots, \theta'_B\}$

    Let $S = \{\}$
    Let $D_0 = D$
    Let $\bar{q}(\theta|x) = \sum_i q_{\phi_i}(\theta'|x)$
    **for** $b \in [1, \ldots, B]$ **do**
        **for** $i \in [1, \ldots, N]$ **do**
            Calculate $M_i = \mathbb{I}[\phi; \theta|x_o, D_{(b-1)} \cup \{(x_o, \theta_i)\}]$
            Let $\tilde{p}_b(\theta_i) = \bar{q}(\theta_i|x_o)\exp\left(-\beta_{\epsilon,N} M_i\right)$         // see Appendix C
        Draw $\tilde{\theta}_b \sim \tilde{p}_b(\theta)$
        Set $D_b = D_{b-1} \cup \{(\tilde{\theta}_b, x_o)\}$
        Set $S = S \cup \{\tilde{\theta}_b\}$
    **return** $S$

---

The shared parameter "suggestion" step takes place when the ensembled proposal prior is constructed pre-simulation, and the collective "wisdom" is disseminated back to each model post-simulation. By Eq. 3 we know that each model converges to the true posterior, but due to variability in $\phi_j$, this may take place at different rates. Figure 3 illustrates this on an example inference run, empirically demonstrating how several underlying models can contribute to a steady rate of convergence while any singular model may converge more slowly on its own.

### 3.2 RESIDUAL MUTUAL INFORMATION

Training with an atomic loss as in APT (Greenberg et al., 2019) can be intuitively likened to "quizzing" the model $q_\phi(\theta|x)$ with multiple choice questions. This is illustrated in Eq. 3: given the $M$ possible options in the batch $\Theta$, the model must learn to correctly assign the most mass to the $(\theta, x)$ pair that was actually observed.

As an extension to this analogy, we posit the following: which questions are the best to ask? Put another way, which questions stand to "teach" our model the most about the target posterior $p(\theta|x_o)$? Here we turn to mutual information as a means of measuring how much we expect to learn from certain observations, and weight the *prospective* impact of those outcomes by how likely the model currently believes them to be. In particular, we want to calculate

$$\mathbb{I}[\phi; \theta|x_o, D'] = \mathbb{E}_{p(\phi|x_o, D')}\left[\mathcal{D}_{KL}\left(p(\theta|x_o, \phi, D')||p(\theta|x_o, D')\right)\right] \tag{8}$$

$$= \mathbb{H}\left[\theta|x_o, D'\right] - \mathbb{E}_{\phi \sim p(\phi|D')}\left[\mathbb{H}\left[\theta|\phi, x_o\right]\right], \tag{9}$$

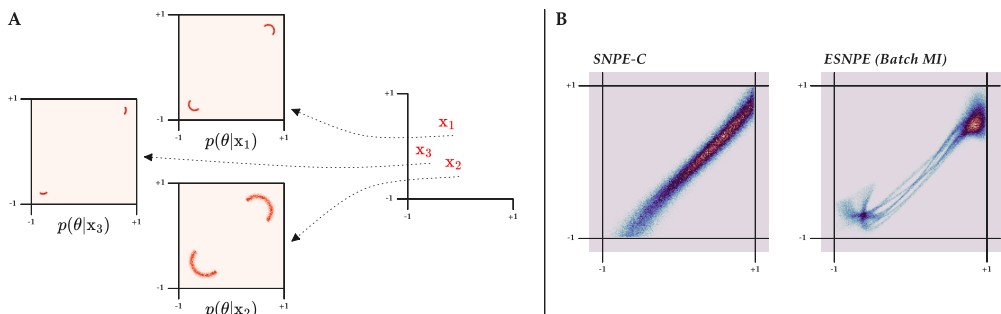

Figure 3: **Posterior plots for the Two Moons task.** **(A)** Two Moons posterior slices under $x_1$, $x_2$, $x_3$, as reported in Table 1. **(B)** Approximate posteriors of $p(\theta|x_1)$ for the Two Moons task after just 128 simulation draws (broken over two rounds of 64 samples).

i.e., the mutual information between simulation parameters $\theta$ and posterior model parameters $\phi$, conditional on observational data of interest $x_o$ and a prospective new dataset $D' = D \cup \{(\theta', x_o)\}$. This term in part mirrors the "active SBI" motivations of Griesemer et al. (2024), but we crucially aim to approximate the mutual information directly as a strategy for sampling guidance (as opposed to strict acquisition optimization of an alternative objective).

Note that $D'$ makes a concrete assumption as to the outcome of any candidate $\theta'$: because we only assume access to an inverse model $q_\phi(\theta|x)$, we cannot generate alternative possible simulation outputs (e.g., $x' \sim p(x|\theta')$; see details regarding expected mutual information in Appendix B). It is in this sense that the approach is considered "residual:" we quantify the remaining information *after* having added the pair to our dataset. We assume $\theta'$ will yield $x_o$, and weight the resulting mutual information estimate according to $q_\phi(\theta'|x_o)$, the current posterior belief of such an outcome. We can therefore naturally re-weight this model before its use as a proposal prior in a manner that preserves the full prior support, e.g.,

$$\tilde{p}(\theta') \propto q_\phi(\theta'|x_o)\exp\left(-\mathbb{I}[\phi; \theta|x_o, D']\right), \tag{10}$$

where we note the negative exponentiation of the mutual information reflects the goal of minimizing this term in the face of our beliefs. This preserves posterior convergence guarantees broadly applicable to atomic proposals, as $\tilde{p}(\theta') > 0$ when $\theta'$ is in the prior support (see Appendix D.2 for details on posterior convergence under proposal reweighting).

A critical practical challenge when calculating this term is the need to leverage only the current ensemble component models $\phi \sim p(\phi|D)$ without performing intermediate retraining for every parameter candidate $\theta'$ (infeasible for even small SBI tasks). We can, however, rewrite Eq. 8 such that it depends only on $p(\phi|D)$ and approximate Eq. 9 accordingly:

$$\mathbb{H}\left[\theta|x_o, D'\right] \approx -\frac{1}{M}\sum_{\theta_i \sim \theta|x_o, D}^{M}\left[\left(\frac{P'_K}{p(\theta'|x_o, D)p(\theta_i|x_o, D)}\right)\log\left(\frac{P'_K}{p(\theta'|x_o, D)}\right)\right] \tag{11}$$

$$\mathbb{E}_{\phi|D'}\left[\mathbb{H}\left[\theta|\phi, x_o\right]\right] \approx -\frac{1}{MK}\sum_{\phi_i \sim \phi|D}^{K}\left[\frac{p(\theta'|x_o, \phi_i)}{p(\theta'|x_o, D)}\sum_{\theta_j \sim \theta|x_o, \phi_i}^{M}\log p(\theta_j|\phi_i, x_o)\right] \tag{12}$$

where $P'_K = \frac{1}{K}\sum_{\phi_j \sim \phi|D}^{K}\left[p(\theta'|x_o, \phi_j)p(\theta_i|x_o, \phi_j)\right]$. See Appendix A.1 for a full derivation and A.2 for implementation details.

### 3.3 BATCH ACQUISITION WITH JOINT MUTUAL INFORMATION

Sequential SBI methods collect simulation data in parallel when possible, drawing a batch of parameters $\theta_1, \ldots, \theta_B \sim \tilde{q}^{(r)}$ from the current round's proposal. The reweighting scheme introduced in Section 3.2 leverages model uncertainty to drive useful parameter exploration, but as with many myopic acquisition schemes, greedily sampling or optimizing to produce a batch can lead to worse-than-random performance, over-representing regions of uncertainty (Kirsch et al., 2019). Batch-aware

Table 1: Results comparing ESNPE variants with baseline methods for C2ST (means and 95% CIs over 10 trials) on *Two Moons*, *Gaussian mixture*, and *Bernoulli GLM* tasks.

| | | Observation | | |
|---|---|---|---|---|
| *Simulator* | *Method* | $p(\theta\|\mathrm{x}_1)$ | $p(\theta\|\mathrm{x}_2)$ | $p(\theta\|\mathrm{x}_3)$ |
| *Two moons* | SMC-ABC | $0.986 \pm 0.001$ | $0.983 \pm 0.001$ | $0.994 \pm 0.001$ |
| | NPSE | $0.905 \pm 0.027$ | $0.872 \pm 0.022$ | $0.956 \pm 0.010$ |
| | FMPE | $0.991 \pm 0.006$ | $0.968 \pm 0.006$ | $0.995 \pm 0.005$ |
| | SNPE-C | $0.898 \pm 0.055$ | $0.869 \pm 0.056$ | $0.927 \pm 0.039$ |
| | TSNPE | $0.948 \pm 0.035$ | $0.929 \pm 0.027$ | $0.967 \pm 0.028$ |
| | ASNPE | $0.815 \pm 0.024$ | $0.846 \pm 0.032$ | $0.903 \pm 0.044$ |
| | ESNPE | $0.804 \pm 0.022$ | $0.802 \pm 0.038$ | $0.871 \pm 0.039$ |
| | ESNPE (MI) | $0.808 \pm 0.027$ | $0.812 \pm 0.029$ | $0.872 \pm 0.025$ |
| | ESNPE (Batch MI) | $\mathbf{0.799 \pm 0.021}$ | $\mathbf{0.798 \pm 0.036}$ | $\mathbf{0.862 \pm 0.016}$ |
| *Gaussian mixture* | SMC-ABC | $0.921 \pm 0.006$ | $0.878 \pm 0.005$ | $0.909 \pm 0.006$ |
| | NPSE | $0.758 \pm 0.004$ | $0.712 \pm 0.007$ | $0.754 \pm 0.014$ |
| | FMPE | $0.914 \pm 0.016$ | $0.882 \pm 0.008$ | $0.891 \pm 0.025$ |
| | SNPE-C | $0.765 \pm 0.049$ | $0.746 \pm 0.050$ | $0.765 \pm 0.081$ |
| | TSNPE | $0.812 \pm 0.039$ | $0.773 \pm 0.021$ | $0.786 \pm 0.010$ |
| | ASNPE | $0.730 \pm 0.005$ | $0.707 \pm 0.013$ | $0.709 \pm 0.008$ |
| | ESNPE | $0.729 \pm 0.006$ | $0.709 \pm 0.011$ | $0.711 \pm 0.012$ |
| | ESNPE (MI) | $0.731 \pm 0.004$ | $0.708 \pm 0.008$ | $\mathbf{0.706 \pm 0.009}$ |
| | ESNPE (Batch MI) | $\mathbf{0.723 \pm 0.014}$ | $\mathbf{0.705 \pm 0.006}$ | $0.712 \pm 0.010$ |
| *Bernoulli GLM* | SMC-ABC | $0.991 \pm 0.001$ | $0.994 \pm 0.000$ | $0.994 \pm 0.001$ |
| | NPSE | $0.917 \pm 0.021$ | $0.924 \pm 0.011$ | $0.915 \pm 0.011$ |
| | FMPE | $0.940 \pm 0.036$ | $0.933 \pm 0.035$ | $0.973 \pm 0.022$ |
| | SNPE-C | $0.793 \pm 0.056$ | $0.802 \pm 0.052$ | $0.755 \pm 0.069$ |
| | TSNPE | $0.966 \pm 0.042$ | $0.965 \pm 0.044$ | $0.926 \pm 0.036$ |
| | ASNPE | $0.823 \pm 0.048$ | $0.809 \pm 0.054$ | $0.731 \pm 0.034$ |
| | ESNPE | $0.774 \pm 0.059$ | $\mathbf{0.765 \pm 0.046}$ | $0.736 \pm 0.052$ |
| | ESNPE (MI) | $0.816 \pm 0.095$ | $0.834 \pm 0.080$ | $\mathbf{0.728 \pm 0.031}$ |
| | ESNPE (Batch MI) | $\mathbf{0.763 \pm 0.054}$ | $0.807 \pm 0.068$ | $0.743 \pm 0.038$ |

acquisition, however, can produce diverse sample sets by jointly optimizing over the entire batch, selecting a coherent collection of samples and reduce information overlap.

We extend the mutual information term $\mathbb{I}[\phi; \theta | x_o, D']$ to allow $D'$ to include arbitrary batches of points, i.e., $D' = D \cup \{(\theta'_i, x_o)\}_{1:B}$. Equations 11 and 12 can be extended accordingly (see Appendix C), but computation of the term for all possible batches quickly becomes infeasible as the candidate pool and $B$ grow. To combat this, we formulate a randomized greedy approximation algorithm for sampling batches with high-information content. Note that deterministically optimizing our batch selection as

$$\{\theta_1^*, \dots, \theta_B^*\} = \mathrm{argmax}_{\{\theta_1, \dots, \theta_B\}} \mathbb{I}[\phi; \theta | x_o, D'] \tag{13}$$

breaks the requirement that the effective proposal $\tilde{p}$ must assign non-zero likelihood to $\theta$ in the prior support. We balance these two objectives – finding jointly informative batches while preserving posterior convergence guarantees – by introducing a stochastic greedy algorithm for selecting high-information batches, yielding a $(1 - 1/e - \epsilon)$-approximation in expectation. See Alg. 2 for a summary of this scheme; Appendix C.2 provides details for approximation guarantees and Appendix D.3 for details on posterior convergence under joint acquisition. Figure 2 depicts the batch selection process in action as samples are collected for inference on the Two Moons task. As points are selected from each (greedily) updated proposal, the conditional model uncertainty changes to reflect previously selected points. For instance, after collecting $\theta'_2$ (a fairly central point in the prior support), the component models become better aligned through the mid-point (fewer disagreeing fibers near $(0, 0)$).

## 4 EXPERIMENTAL RESULTS

To quantify the impact of the proposed ESNPE scheme and its active variants, we evaluate our method on five SBI benchmark tasks Lueckmann et al. (2021) and compare against several existing NPE-adjacent approaches. Each task presents with samples from several ground-truth posterior slices $p(\theta | \mathrm{x}_i)$, allowing for precise evaluation of posterior fit. We primarily report the classifier

two-sample test (C2ST) accuracy between ground-truth and approximate samples; a score of 0.5 indicates the classifier finds the two samples indistinguishable, whereas a score of 1.0 indicates the samples originate from distinctly different distributions.

## 4.1 BASELINE NPE METHODS

Benchmark tasks are evaluated on several available NPE baseline methods. SMC-Approximate Bayesian computation (SMC-ABC) Rubin (1984); Beaumont et al. (2002); Sisson et al. (2007); Marjoram et al. (2003) is the only non-neural network-based method. Flow-matching Dax et al. (2023), and score estimation Geffner et al. (2023b) estimation are considered as they directly approximate the posterior, but their formulations do not support multi-round inference under flexible proposals, and instead draw their entire simulation budget from the prior. SNPE-C Greenberg et al. (2019) and ASNPE Griesemer et al. (2024) are both multi-round inference procedures that leverage atomic losses, and therefore have the most in common with our method. TSNPE (Deistler et al., 2022) is also a multi-round inference method, but it explicitly truncates the prior in producing the posterior approximation. We report results for the non-active ESNPE scheme, as well as the single MI and batch MI acquisition variants introduced in Sections 3.2 and 3.3, respectively.

## 4.2 COMMON SBI BENCHMARKS

Table 1 reports C2ST scores for each baseline method on three common SBI benchmark tasks: Two Moons, Gaussian mixture, and Bernoulli GLM Lueckmann et al. (2021). These tasks vary in difficulty for direct posterior estimation methods. For each task, we evaluate the baseline methods on three different posterior slices to capture more breadth under the simulation dynamics (some $x_i$ yield far simpler posteriors, for instance). The simulation budget for each setting is fixed at 1,024 samples, with multi-round methods carrying out inference across four rounds of 256 samples. Across every task and observation, ESNPE-based methods report the best C2ST scores, with the Batch MI acquisition generally yielding a larger performance advantage. Figure 3 visualizes each of the evaluated posterior slices, and highlights posteriors from SNPE-C and ESNPE on an example Two Moons inference round. In the latter case, the posterior plot shows the individual component models as they seek to find a consistent representation of the data; in low-data settings, component posteriors can disagree heavily, as seen with the many "tendrils" stemming from a unified mode in the top right corner.

## 4.3 DIFFERENTIAL EQUATION-BASED ENVIRONMENTS

We additionally evaluate ESNPE on the more challenging Lotka-Volterra and SIR tasks, both of which have simulation dynamics described by differential equations and produce time series observations.

**SIR model**: The SIR (susceptible-infected-removed) model captures simple disease spreading dynamics in human populations. Its dynamics are governed by two parameters: a contact rate $\beta$ and recovery rate $\gamma$. Simulations produce the number of infected individuals as a 10-sample time series in the evolution of the spreading disease across the population.

Table 2: C2ST and MMD scores for SNPE-C and ESNPE on the Lotka-Volterra and SIR tasks.

| Task | Obs. | Method | Metric | |
| --- | --- | --- | --- | --- |
| | | | C2ST | MMD |
| Lotka Volterra | $\mathbf{x}_1$ | SNPE-C | $0.94 \pm 0.01$ | $0.47 \pm 0.01$ |
| | | ESNPE | $\mathbf{0.91 \pm 0.02}$ | $\mathbf{0.33 \pm 0.01}$ |
| | $\mathbf{x}_2$ | SNPE-C | $0.89 \pm 0.03$ | $0.30 \pm 0.17$ |
| | | ESNPE | $\mathbf{0.87 \pm 0.02}$ | $\mathbf{0.29 \pm 0.03}$ |
| SIR model | $\mathbf{x}_1$ | SNPE-C | $0.91 \pm 0.10$ | $\mathbf{0.87 \pm 0.35}$ |
| | | ESNPE | $\mathbf{0.89 \pm 0.13}$ | $0.88 \pm 0.52$ |
| | $\mathbf{x}_2$ | SNPE-C | $0.66 \pm 0.12$ | $0.14 \pm 0.15$ |
| | | ESNPE | $\mathbf{0.57 \pm 0.02}$ | $\mathbf{0.01 \pm 0.01}$ |

**Lotka-Volterra**: Lotka-Volterra Wangersky (1978); Lotka (1927) is a classic predator-prey model in ecology. Its dynamics are determined by four parameters governing birth and death rates in the population, and observations are drawn over a sequence of 20 days.

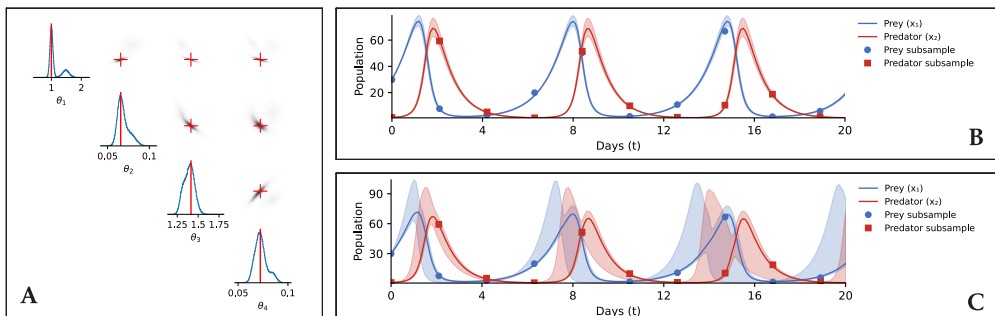

Figure 4: **Posterior plots for the Lotka-Volterra task.** **(a)** Approximate posterior slice $p(\theta|\mathrm{x}_2)$ learned by ESNPE, visualized as pairplots in the 4-D parameter space. **(b)** Noise in the output space as seen by samples $x_i \sim p(x|\theta)$ for $\theta \sim p(\theta|\mathrm{x}_2)$. **(c)** Noise in the output space under the approximate posterior, i.e., $x_i \sim p(x|\theta)$ for $\theta \sim q_\Phi(\theta|\mathrm{x}_2)$. The red plus in subplot (a) corresponds to the posterior MAP, and fill area corresponds to a 5%-95% envelope around the median path. The discrete subsample points represent the 20-dimensional reference observation $\mathrm{x}_2$.

In Table 2, we report C2ST and maximum mean discrepancy (MMD) scores for two observations across both tasks. Here we highlight the SNPE-C and ESNPE methods in order to isolate the effects of the ensemble dynamics. Both models are run for eight inference rounds, collecting 256 samples per round. We observe that ESNPE attains a consistent performance advantage over SNPE-C in each setting and on both metrics, with the exception of the MMD score under $\mathrm{x}_1$ of the SIR model.

Figure 4 shows reference samples for the Lotka-Volterra simulator. Subplot (c) shows several draws from the posterior approximation learned by ESNPE under observation $\mathrm{x}_2$. While the reference observation (the discrete subsample points) is covered under the posterior draws, there are distinct indications of mis-calibrated posterior mass that show in the observation space (e.g., $t \approx 8$, $t \approx 14$ for the predator sample).

The results seen across common SBI benchmarks, as well as the more challenging differential equation-based settings, indicate ESNPE's broad performance advantage with respect to key posterior quality metrics like C2ST and MMD. The dependence of ESNPE on several underlying models can increase the computational cost, however, especially when parallelization is difficult. Please see Appendix F for additional in-depth analyses, including a discussion on differences across batched and non-batched method performance, time complexity and empirical runtimes, etc. Appendix G provides additional experimental results and hyperparameter ablations that may help guide practitioners when evaluating the compute-to-performance tradeoffs.

## 5 CONCLUSION

Recent advances in SBI methods have significantly improved the feasibility of performing accurate likelihood-free inference in challenging real-world settings. Operating in low-data settings remains a core practical consideration, however, especially when working with slow/expensive simulation systems. In this work, we proposed a general ensembling scheme for collectively training and combining groups of atomic proposal priors across rounds of sequential posterior estimation. We extended this scheme by proposing a means of approximating the information content of prospective simulation parameter candidates, further capitalizing on model uncertainty and guiding sampling toward useful regions of the parameter space. We extended this scheme to the batch setting and formulated a fast, greedy approximation algorithm that retains posterior convergence guarantees while targeting informative batches of simulation parameters. Across several common SBI benchmarks, we observe the ensemble schemes outperform baseline methods on relevant measures of posterior quality when operating under tight simulation budgets. These results indicate carefully balanced ensemble methods with active selection schemes are strong contenders for tackling challenging SBI tasks in a more stable, sample-efficient manner than existing approaches.

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

# Appendix

# A  MUTUAL INFORMATION UNDER PROSPECTIVE DATA

In Section 3.2, we formulate a notion of residual mutual information where we seek to quantify the remaining information after having observed individual parameter prospects. Fully calculating this term requires entire ensemble updates (as discussed in Section E), which can be computationally expensive for even moderately sized $K$ or parameter pools.

## A.1  DEPENDENCE UNDER $p(\phi|D)$

In this section, we decompose the mutual information as presented in Eq. 8 to arrive at a means of approximating entropy terms when we only permit a conditional dependence on the current dataset $D$ (i.e., assuming we can't explicitly perform model updates under every candidate $\theta'$ in practice).

Let $D' = D \cup \{(\theta', x_o)\}$. Then we have the mutual information between inference parameters $\theta$ and posterior model parameters $\phi$:

$$
\begin{aligned}
\mathbb{I}[\phi; \theta | x_o, D'] &= \int_\Phi \int_\Theta p(\theta, \phi | x_o, D') \log \left( \frac{p(\theta, \phi | x_o, D')}{p(\theta | x_o, D')p(\phi | x_o, D')} \right) d\theta d\phi \\
&= \int_\Phi \int_\Theta p(\theta | \phi, x_o, D')p(\phi | x_o, D') \log \left( \frac{p(\theta | \phi, x_o, D')p(\phi | x_o, D')}{p(\theta | x_o, D')p(\phi | x_o, D')} \right) d\theta d\phi \\
&= \int_\Phi p(\phi | x_o, D') \left[ \int_\Theta p(\theta | \phi, x_o, D') \log \left( \frac{p(\theta | \phi, x_o, D')}{p(\theta | x_o, D')} \right) d\theta \right] d\phi \\
&= \int_\Phi p(\phi | x_o, D') \left[ \mathcal{D}_{KL} \left( p(\theta | \phi, x_o, D') || p(\theta | x_o, D') \right) \right] d\phi \\
&= \mathbb{E}_{p(\phi | x_o, D')} \left[ \mathcal{D}_{KL} \left( p(\theta | \phi, x_o, D') || p(\theta | x_o, D') \right) \right].
\end{aligned}
\tag{14}
$$

The mutual information can also be expressed in terms of entropies:

$$
\begin{aligned}
\mathbb{I}[\phi; \theta | x_o, D'] &= \int_\Theta \int_\Phi p(\phi, \theta | x_o, D') \log \left( \frac{p(\phi, \theta | x_o, D')}{p(\phi | x_o, D')p(\theta | x_o, D')} \right) d\phi d\theta \\
&= \int_\Theta \int_\Phi p(\phi, \theta | x_o, D') \log \left( \frac{p(\phi | \theta, x_o, D')p(\theta | x_o, D')}{p(\phi | x_o, D')p(\theta | x_o, D')} \right) d\phi d\theta \\
&= \int_\Theta \int_\Phi p(\phi, \theta | x_o, D') \log p(\phi | \theta, x_o, D') d\phi d\theta \\
&\quad - \int_\Theta \int_\Phi p(\phi, \theta | x_o, D') \log p(\phi | x_o, D') d\phi d\theta \\
&= -H \left[ \phi | \theta, x_o, D' \right] - \int_\Theta \int_\Theta p(\theta | \phi, x_o, D')p(\phi | x_o, D') \log p(\phi | x_o, D') d\phi d\theta \\
&= -H \left[ \phi | \theta, x_o, D' \right] - \int_\Theta p(\phi | x_o, D') \log p(\phi | x_o, D') \left[ \int_\Theta p(\theta | \phi, x_o, D') d\theta \right] d\phi \\
&= -\mathbb{H} \left[ \phi | \theta, x_o, D' \right] + \mathbb{H} \left[ \phi | x_o, D' \right] \\
&= \mathbb{H} \left[ \phi | x_o, D' \right] - \mathbb{E}_{\theta \sim p(\theta | D')} \left[ \mathbb{H} \left[ \phi | \theta, x_o \right] \right],
\end{aligned}
\tag{15}
$$

and by symmetry of the joint factorization, we have generally

$$
\begin{aligned}
\mathbb{I}[\phi; \theta | x_o, D'] &= \mathbb{E}_{p(\theta | x_o, D')} \left[ \mathcal{D}_{KL} \left( p(\phi | \theta, x_o, D') || p(\phi | x_o, D') \right) \right] \\
&= \mathbb{E}_{p(\phi | x_o, D')} \left[ \mathcal{D}_{KL} \left( p(\theta | \phi, x_o, D') || p(\theta | x_o, D') \right) \right] \\
&= \mathbb{H} \left[ \theta | x_o, D' \right] - \mathbb{E}_{\phi \sim p(\phi | D')} \left[ \mathbb{H} \left[ \theta | \phi, x_o \right] \right] \\
&= \mathbb{H} \left[ \phi | x_o, D' \right] - \mathbb{E}_{\theta \sim p(\theta | D')} \left[ \mathbb{H} \left[ \phi | \theta, x_o \right] \right].
\end{aligned}
\tag{16}
$$

Out of convenience, we seek to approximate the form

$$
\mathbb{I}[\phi; \theta | x_o, D'] = \mathbb{H} \left[ \theta | x_o, D' \right] - \mathbb{E}_{\phi \sim p(\phi | D')} \left[ \mathbb{H} \left[ \theta | \phi, x_o \right] \right].
\tag{17}
$$

We first note that, by Bayes' rule,

$$p(\phi|D') = \frac{p(\theta'|x_o,\phi)p(\phi|D)}{p(\theta'|x_o,D)}, \tag{18}$$

which can be used to rewrite a posterior dependence on $D'$ in terms of $D$:

$$\begin{aligned}
p(\theta|x_o,D') &= \int p(\theta|x_o,\phi)p(\phi|D')d\phi \\
&= \int p(\theta|x_o,\phi)\left[\frac{p(\theta'|x_o,\phi)p(\phi|D)}{p(\theta'|x_o,D)}\right]d\phi \\
&= \frac{1}{p(\theta'|x_o,D)}\int p(\theta|x_o,\phi)p(\theta'|x_o,\phi)p(\phi|D)d\phi \\
&= \frac{1}{p(\theta'|x_o,D)}\mathbb{E}_{\phi|D}\left[p(\theta|x_o,\phi)p(\theta'|x_o,\phi)\right],
\end{aligned} \tag{19}$$

The entropy terms can be rewritten accordingly, yielding a representation of $\mathbb{I}[\phi;\theta|x_o,D']$ when we just have access to a model $p(\theta|x_o,D)$ and model weight posterior $p(\phi|D)$ trained up to data $D$ (rather than $D'$ explicitly). For the "marginal entropy":

$$\begin{aligned}
\mathbb{H}[\theta|x_o,D'] &= \mathbb{E}_{\theta|x_o,D'}\left[-\log p(\theta|x_o,D')\right] \\
&= -\mathbb{E}_{\theta|x_o,D}\left[\left(\frac{p(\theta|x_o,D')}{p(\theta|x_o,D)}\right)\log p(\theta|x_o,D')\right] \\
&= -\mathbb{E}_{\theta|x_o,D}\left[\left(\frac{\mathbb{E}_{\phi|D}\left[p(\theta'|x_o,\phi)p(\theta|x_o,\phi)\right]}{p(\theta'|x_o,D)p(\theta|x_o,D)}\right)\log\left(\frac{\mathbb{E}_{\phi|D}\left[p(\theta'|x_o,\phi)p(\theta|x_o,\phi)\right]}{p(\theta'|x_o,D)}\right)\right] \\
&= -\mathbb{E}_{\theta|x_o,D}\left[\left(\frac{P'_{K,\theta}}{p(\theta'|x_o,D)p(\theta|x_o,D)}\right)\log\left(\frac{P'_{K,\theta}}{p(\theta'|x_o,D)}\right)\right]
\end{aligned} \tag{20}$$

where

$$P'_{K,\theta} = \frac{1}{K}\sum_{\phi_j\sim\phi|D}^{K}\left[p(\theta'|x_o,\phi_j)p(\theta|x_o,\phi_j)\right], \tag{21}$$

and the "expected conditional entropy"

$$\begin{aligned}
\mathbb{E}_{\phi|D'}\left[\mathbb{H}[\theta|\phi,x_o]\right] &= \mathbb{E}_{\phi|D'}\left[\mathbb{E}_{\theta|x_o,\phi}\left[-\log p(\theta|\phi,x_o)\right]\right] \\
&= \mathbb{E}_{\phi|D}\left[\frac{p(\theta'|x_o,\phi)}{p(\theta'|x_o,D)}\mathbb{E}_{\theta|x_o,\phi}\left[-\log p(\theta|\phi,x_o)\right]\right].
\end{aligned} \tag{22}$$

A.2 EFFICIENT APPROXIMATION

With the mutual information term represented in $\phi|D$, expectations can be approximated via an MC estimate over $K$ *current* ensemble components, i.e.,

$$\mathbb{H}[\theta|x_o,D'] \approx -\frac{1}{M}\sum_{\theta_i\sim\theta|x_o,D}^{M}\left[\left(\frac{P'_{K,\theta_i}}{p(\theta'|x_o,D)p(\theta_i|x_o,D)}\right)\log\left(\frac{P'_{K,\theta_i}}{p(\theta'|x_o,D)}\right)\right] \tag{23}$$

and

$$\mathbb{E}_{\phi|D'}\left[\mathbb{H}[\theta|\phi,x_o]\right] \approx -\frac{1}{LK}\sum_{\phi_i\sim\phi|D}^{K}\left[\frac{p(\theta'|x_o,\phi_i)}{p(\theta'|x_o,D)}\sum_{\theta_j\sim\theta|x_o,\phi_i}^{L}\log p(\theta_j|\phi_i,x_o)\right] \tag{24}$$

These terms can be calculated efficiently over a candidate pool of size $N$. We define the following matrices:

- Matrix $P'_{NK}$ shaped $N \times K$, where $(P'_{NK})_{n,k} = p(\theta'_n | x_o, \phi_k)$ for the $n$-th candidate parameter $\theta'_n$ and $k$-th posterior model $\phi_k$.

- Matrix $P_{MK}$ shaped $M \times K$, where $(P_{MK})_{m,k} = p(\theta_m | x_o, \phi_k)$, and $\theta_m \sim p(\cdot | x_o, D)$. Note that $\theta_m$ are indeed drawn from the entire ensemble (which is composed of individual $\phi_k$), but the stored values include each of the $K$ models' evaluated likelihoods on those parameters.

- Matrix $P_{LK}$ shaped $L \times K$, where $(P_{LK})_{\ell,k} = p(\theta_\ell | x_o, \phi_k)$, and $\theta_\ell \sim p(\cdot | x_o, \phi_k)$ for $k$-th posterior model $\phi_k$. Here the parameters evaluated under each model are *not* shared across component models but drawn and evaluated under each model individually.

These matrices can be computed upfront for each round of inference, given the current set of ensemble components $\{\phi_1, \ldots, \phi_K\}$ and parameter candidate pool $\{\theta'_1, \ldots, \theta'_N\}$. The necessary quantities for the entropy approximations can then be calculated efficiently in a vectorized manner:

- Parameter candidate marginals: $p(\theta' | x_o, D) = \frac{1}{K} P'_{NK} \mathbb{1}_{K,1}$

- Ensemble marginals: $p(\theta | x_o, D) = \frac{1}{K} P_{MK} \mathbb{1}_{K,1}$

- Component log-likelihood sums: $\sum_{\theta_j \sim \theta | x_o, \phi_i}^{L} \log p(\theta_j | \phi_i, x_o) = P_{LK}^{\top} \mathbb{1}_{L,1}$

- Expected component joint likelihood: $P'_K = \frac{1}{K} P_{NK} P_{MK}^{\top}$, where

$$P'_{K,i} = \frac{1}{K} \sum_{\phi_j \sim \phi | D}^{K} [p(\theta' | x_o, \phi_j) p(\theta_i | x_o, \phi_j)].$$

where $\mathbb{1}_{K,1}$ is a $K$-sized column vector of 1s. This structure is amenable to a dynamic programming approach for larger batch sizes, and can be trivially expanded under the routine in Alg. 2 and detailed further in Appendix C. See further runtime analyses in Appendix F.3.

## B   LIKELIHOOD SURROGATE EXTENSION

The prospective mutual information terms discussed in Sections 3.2 and C explicitly consider the impact of observing $x_o$ for parameter candidates. This is a convenient option provided it only requires an inverse model (we can draw likely $\theta' \sim p(\theta | x_o)$ directly), but this perspective can yield a shallow estimate given it only incorporates a single output target. In theory our prospective terms could include an outer expectation over likely parameter outputs for each candidates, i.e.,

$$\max_{\theta' \sim \theta | x_o} \mathbb{E}_{x' \sim x | \theta'} [\mathbb{I}[\phi; \theta | x_o, D \cup \{(\theta', x')\}]].$$

While such a scheme may yield a more holistic estimate of a prospective parameter's impact, we avoid this form due to its reliance on an independent forward model $p(x' | \theta')$ which can run counter to the uncertainty present across the direct posterior models being used to produce round-wise proposals.

## C   BATCH ACQUISITION

### C.1   PROSPECTIVE BATCH FORMULATION

Section A formulates the mutual information term around a prospective dataset $D' = D \cup \{(\theta', x_o)\}$, i.e., only considering the impact of a single pair $(\theta', x_o)$. In the batched setting, however, we want to consider the impact of an entire collection of points $\{(\theta_i, x_i)\}_{1:B}$. This enables the selection method to better coordinate the collective impact of the simulation points being considered for the next inference round.

Here we consider a similar formulation to that used in Section A, but include joint probabilities over several candidates under $D' = D \cup \{(\theta_i, x_i)\}_{1:B}$:

$$p(\phi | D') = \frac{\prod_i [p(\theta'_i | x'_i, \phi)] p(\phi | D)}{\prod_i p(\theta'_i | x'_i, D)}, \tag{25}$$

with the associated expansion of the marginal $p(\theta|x_o, D')$:

$$
\begin{aligned}
p(\theta|x_o, D') &= \int p(\theta|x_o, \phi)p(\phi|D')d\phi \\
&= \int p(\theta|x_o, \phi) \left[ \frac{\prod_i \left[ p(\theta'_i|x'_i, \phi) \right] p(\phi|D)}{\prod_i \left[ p(\theta'_i|x'_i, D) \right]} \right] d\phi \\
&= \frac{1}{\prod_i \left[ p(\theta'_i|x'_i, D) \right]} \mathbb{E}_{\phi|D} \left[ \prod_i p(\theta'_i|x'_i, \phi) \right].
\end{aligned}
\tag{26}
$$

Finding the optimal batch over the associated mutual information term term, i.e.,

$$
\{\theta_1^*, \ldots, \theta_B^*\} = \underset{\{\theta_1, \ldots, \theta_B\} \subseteq \Theta'}{\arg\min} \ \mathbb{H}\left[\theta|x_o, D'\right] - \mathbb{E}_{\phi \sim p(\phi|D')}\left[\mathbb{H}\left[\theta|\phi, x_o\right]\right],
\tag{27}
$$

where $\Theta'$ is an $N > B$-sized set of candidate parameters, is computationally infeasible due to the combinatorially large number of batches to consider. As in Section C, we also consider constraints when only an inverse model is available, namely drawing candidates from the current posterior $\theta' \sim q_\phi(\theta|x_o)$ and considering the impact under $x_o$ with post-hoc reweighting.

## C.2 RANDOM GREEDY $(1 - 1/e - \epsilon)$-APPROXIMATION

The greedy approximation procedure described in Alg. 2 seeks to optimize the joint mutual information of the selected batch, but with two concessions: 1) an efficient, greedy approximation, and 2) randomized selection to preserve prior support. Below we show the former yields a $(1 - 1/e)$-approximate algorithm via submodularity, and relax this to a $(1 - 1/e - \epsilon)$ approximation in expectation to meet the latter.

### C.2.1 PROOF OF SUBMODULARITY

Nemhauser et al. Nemhauser et al. (1978) provide an analytical basis for simple greedy approximations when maximizing a monotone submodular set function. Submodularity captures a notion of diminishing returns, which intuitively applies when optimizing mutual information-based terms point-by-point.

**Definition C.1.** Let $f$ be a real-valued function defined on subsets of a universe $\mathcal{U}$. $f$ is submodular if, for all $S \subseteq T \subseteq \mathcal{U}$ and $u \notin T$,

$$
f(S \cup \{u\}) - f(S) \geq f(T \cup \{u\}) - f(T)
\tag{28}
$$

Let $\mathbb{M}(\Theta) = \mathbb{I}[\phi; \theta|x_o, D \cup \{(\theta', x_o)\}_{\theta' \in \Theta}]$, a real-valued set function representing the batch-wise mutual information under a batch of parameters $\Theta \subseteq \Theta'$, where $\Theta'$ is a fixed set of candidates (see also Eq. 27).

**Lemma C.1.** Let $\Theta_S = \{i \in S|\Theta_i\}$. The marginal difference in $\mathbb{M}$ under the additional simulation parameter $\theta_u$ is

$$
\Delta_u(\Theta_S) = \mathbb{M}(\Theta_S \cup \{\theta_u\}) - \mathbb{M}(\Theta_S) = -\mathbb{I}[\theta_u; \theta|x_o, D, \Theta_S].
\tag{29}
$$

*Proof.* Given the existing batch selection $\Theta_S$ and an additional simulation parameter $\theta_s$, the marginal difference under $\mathbb{M}$ can be expanded as

$$
\mathbb{M}(\Theta_S \cup \{\theta_u\}) - \mathbb{M}(\Theta_S) = \mathbb{I}[\phi; \theta|x_o, D, \Theta_S, \theta_u] - \mathbb{I}[\phi; \theta|x_o, D, \Theta_S]
\tag{30}
$$

By definition, we have the *information interaction*

$$
\mathcal{I}\left(\phi; \theta; \theta_u|x_o, D, \Theta_S\right) = \mathbb{I}[\phi; \theta|x_o, D, \Theta_S] - \mathbb{I}[\phi; \theta|x_o, D, \Theta_S, \theta_u]
\tag{31}
$$

Note that the information interaction is symmetric for any random triple $(X, Y, Z)$

$$
\mathcal{I}(X; Y; Z|W) = \mathbb{I}[Y; Z|W] - \mathbb{I}[Y; Z|W, X].
\tag{32}
$$

We therefore write the information interaction $\mathcal{I}(\phi;\theta;\theta_u|\cdot)$ as

$$
\begin{aligned}
\mathcal{I}(\phi;\theta;\theta_u|x_o,D,\Theta_S) &= \mathbb{I}[\phi;\theta|x_o,D,\Theta_S] - \mathbb{I}[\phi;\theta|x_o,D,\Theta_S,\theta_u] \\
&= \mathbb{I}[\theta;\theta_u|x_o,D,\Theta_S] - \mathbb{I}[\theta;\theta_u|x_o,\phi,D,\Theta_S]
\end{aligned}
\tag{33}
$$

Since considered parameters $\theta_u$ are drawn $\theta_u \sim p(\theta|x_o,\phi), \phi \sim p(\phi|D)$, we have conditional independence in the rightmost mutual information term above, and thus $\mathbb{I}[\theta;\theta_u|x_o,\phi,D,\Theta_S] = 0$. In total, this yields

$$
\begin{aligned}
\Delta_u(\Theta_S) &= \mathbb{M}(\Theta_S \cup \{\theta_u\}) - \mathbb{M}(\Theta_S) \\
&= \mathbb{I}[\phi;\theta|x_o,D,\Theta_S,\theta_u] - \mathbb{I}[\phi;\theta|x_o,D,\Theta_S] \\
&= -\mathcal{I}(\phi;\theta;\theta_u|x_o,D,\Theta_S) \\
&= -\mathbb{I}[\theta;\theta_u|x_o,D,\Theta_S].
\end{aligned}
\tag{34}
$$

$\square$

**Lemma C.2.** $\mathbb{M}$ is submodular over sets $\Theta \subseteq \Theta'$.

*Proof.* Let $\Theta_S \subseteq \Theta_T \subseteq \Theta'$. For $\theta_u \notin \Theta_T$,

$$
\mathbb{I}[\theta;\theta_u|x_o,D,\Theta_S] \leq \mathbb{I}[\theta;\theta_u|x_o,D,\Theta_T]
\tag{35}
$$

which follows from the general "information never hurts" principle Krause & Guestrin (2012) $\mathbb{I}[X;Y|Z,W] \leq \mathbb{I}[X;Y|Z]$ for any random variables $X,Y,Z,W$. This is equivalent to

$$
\begin{aligned}
\Delta_u(\Theta_S) &\geq \Delta_u(\Theta_T) \\
\implies \mathbb{M}(\Theta_S \cup \{\theta_u\}) - \mathbb{M}(\Theta_S) &\geq \mathbb{M}(\Theta_T \cup \{\theta_u\}) - \mathbb{M}(\Theta_T)
\end{aligned}
\tag{36}
$$

using the form derived in lemma C.1. This is exactly the desired submodularity inequality from Eq. 28; hence $\mathbb{M}$ is submodular over sets $\Theta \subseteq \Theta'$. $\square$

Note further that the (conditional) mutual information is non-negative and thus $\Delta_u(\Theta_S) \leq 0$, ensuring the objective is monotone non-increasing. Hence our objective is monotonic and submodular; by Nemhauser et al. Nemhauser et al. (1978), greedy selection of candidates under $\mathbb{M}$ yields a $(1-1/e)$-approximate algorithm for the optimal batch.

### C.2.2 STOCHASTIC RELAXATION

Deterministic optimization via a greedy $(1-1/e)$-approximate routine as shown in Section C.1 can violate posterior convergence guarantees by effectively assigning zero likelihood to parameters in the support of the prior. Here we preserve the greedy framework to loosely uphold the optimal batch approximation, but introduce a random sampling step to ensure all candidates in the prior support have some possibility of selection (with an $\epsilon$-factor concession).

Nemhauser et al. (Nemhauser et al., 1978) further show that if one greedily selects items with marginal gain at least $(1-\epsilon)\Delta_{\max(S)}$, then after $B$ steps $f(S_B) \geq (1-1/e-\epsilon)f(\text{OPT}_B)$, where $\text{OPT}_B$ is the maximal $B$-sized set under $f$. We seek to show our scheme attains this in *expectation* at each step, i.e.,

$$
\mathbb{E}[\Delta_{i_t}(S_{t-1})] \geq (1-\epsilon)\max_{j \notin S_{t-1}}\Delta_j(S_{t-1}).
\tag{37}
$$

We say the resulting scheme attains a $(1-1/e-\epsilon)$-approximation in expectation.

**Lemma C.3.** Let $\beta_{\epsilon,N}(M_{b_i}) = \alpha_{\epsilon,N}(M_{b_i} - M_{b-1}) = \alpha_{\epsilon,N}\Delta_{b_i}(\Theta_{b-1})$, where $M_{b_i} = \mathbb{I}[\phi;\theta|x_o,D_{b-1}\cup\{(x_o,\theta_{b_i})\}]$, $\theta_{b_i}$ is the $i$-th candidate being considered at step $b$, and $M_{b-1}$ is the MI term for the candidate selected in step $b-1$. Then

$$
\mathbb{E}[\Delta(\Theta_{b-1})] \geq (1-\epsilon)\max_{\theta_j \notin \Theta_{b-1}}\Delta_j(\Theta_{b-1})
\tag{38}
$$

when $\tilde{p}_b(\theta_{b_i}) \propto \bar{q}(\theta_i|x_o)\exp(-\beta_{\epsilon,N}(M_{b_i}))$ (from Alg. 2) and $\alpha_{\epsilon,N} \geq \frac{\log N}{\epsilon\Delta_{\max}}$.

*Proof.* Expanding, we have

$$\tilde{p}_b(\theta_{b_i}) = \frac{\exp\left(\alpha_{\epsilon,N}\Delta_{b_i}(\Theta_{b-1})\right)}{\sum_j \exp\left(\alpha_{\epsilon,N}\Delta_{b_j}(\Theta_{b-1})\right)} \tag{39}$$

(note that we temporarily ignore $\bar{q}(\theta_i|x_o)$, as it's a post-hoc re-weight given the assumed output).

Then the expectation

$$\mathbb{E}_{\tilde{p}_b}[\Delta(\Theta_{b-1})] = \sum_{\theta_j \notin \Theta_{b-1}} \tilde{p}_{b_j}\Delta_{b_j}(\Theta_{b-1}) = \frac{L}{\alpha_{\epsilon,N}} - \frac{H[p]}{\alpha_{\epsilon,N}} \tag{40}$$

where $L = \log\sum_j \exp\left(\alpha_{\epsilon,N}\Delta_{b_j}(\Theta_{b-1})\right)$. We then bound each term in the expanded expectation, with $H[p] \leq \log N$ (by entropy convexity) and $L \geq \alpha_{\epsilon,N}\Delta_{\max}$ (by definition of $L$). Hence

$$\mathbb{E}_{\tilde{p}_b}[\Delta(\Theta_{b-1})] \geq \Delta_{\max} - \frac{\log N}{\alpha_{\epsilon,N}}, \tag{41}$$

and when $\alpha_{\epsilon,N} \geq \frac{\log N}{\epsilon\Delta_{\max}}$, plugging in yields $\mathbb{E}_{\tilde{p}_b}[\Delta(\Theta_{b-1})] \geq (1-\epsilon)\Delta_{\max}$. $\qquad\square$

# D   POSTERIOR CONVERGENCE

In Sections 3.1, 3.2, and 3.3, we introduce changes to otherwise valid atomic proposal distributions. Here we show that these modifications yield valid atomic proposals or otherwise observe necessary posterior convergence guarantees.

## D.1   ATOMIC ENSEMBLE PROPOSALS

In ESNPE (Algorithm 1), we ensemble $K$ atomic proposal distributions produced under the APT loss scheme, as shown in Eq. 3. In particular, the round-wise proposal is defined as

$$\bar{q}_\Phi^{(r)}(\theta|x) = \sum_{j=1}^K \tilde{w}_j^{(r)} q_{\phi_j}(\theta|x), \tag{42}$$

which is merely a linear combination of atomic proposals. After the round's parameters $\Theta = \{\theta_1, \ldots, \theta_B\}$ are drawn i.i.d. from the proposal distribution, the loss scheme (as shown in Eq. 4) no longer depends on the form of the proposal distribution itself. Further, each component distribution in the ensemble is an individually valid proposal (it's part of its own APT trajectory), so each component assigns non-zero likelihoods to parameters in the prior support. Therefore any setting of $\tilde{w}^{(r)}$, including the extreme case where a single component receives all of the weight (recall that $\tilde{w}$ is self-normalized), yields an ensemble that is a valid atomic proposal covering the entire prior support.

## D.2   WEIGHTED RESAMPLING

As highlighted in Sections 2.2 and D.1, APT's atomic loss scheme does not explicitly depend on the form of the proposal distribution once a batch of parameters $\Theta$ has been drawn. Hence any reweighting of a valid proposal remains valid, so long as it upholds the same support. Let $w : \Theta \to \mathbb{R}$ be an arbitrary real-valued weighting function defined over the parameter space $\Theta$. Then $\tilde{p}_w(\theta) \propto \tilde{p}(\theta)\exp(w(\theta))$ is a valid atomic proposal when $\tilde{p}$ is, as $\exp(w(\theta)) > 0$ for all $\theta \in \Theta$.

## D.3   JOINT ACQUISITION

Proposition 1 of Greenberg et al. (2019) shows that the APT loss scheme recovers the full posterior shape in the limit as $N \to \infty$ in expectation under data collected $\Theta \sim V, \theta \sim U_\Theta, x \sim p(x|\theta)$, where $V(\Theta)$ is a joint "hyperproposal" defined over parameter batches $\Theta$. $V$ is generally allowed to range over parameter batches that include data drawn from previous rounds, and therefore collected under different point-wise proposals. The batch scheme introduced in Alg. 2 can simply be seen as factorizing a particular choice of $V$ (where each factor is atomic and covering the prior support) at each round, where batches $\Theta \sim V$ are drawn collectively, and as seen in the standard APT scheme can range over previously drawn $\theta$ from arbitrary atomic proposals.

**Algorithm 3** ENSPE update step (ensemble rejuvenation)

---

**Input:** Unweighted sample $\phi_1, \ldots, \phi_K \sim p(\phi|D)$, batch size $B$, candidate parameters $\{\theta_1, \ldots, \theta_N\}$, current round $r$, dataset $D^{(r)}$
**Output:** Rejuvenated, unweighted ensemble $\hat{\phi}_1, \ldots, \hat{\phi}_K$

    [Re-weight] Compute weights $w_i^{(r)}$ for $i = 1, \ldots, K$

    [Re-sample] Draw $\hat{\phi}_i \sim \text{Cat}\left(w_1^{(r)}, \ldots, w_K^{(r)}\right)$

    [Re-train] Run $S$ SGD steps under each component $q_{\hat{\phi}_i}$ via WLB under $D^{(r)}$

---

# E  SAMPLING FROM $p(\phi|D)$

In Section 3.1, we alluded to more in-depth strategies for rejuvenating ensemble components beyond simply re-weighting component contributions. Here we include additional details for a routine that updates posterior models in a particle filtering-like manner. Note that this scheme can be seen as an optional extension to the re-weighting scheme presented in Section 3.1 in terms of theoretical implications, but in practice it may have a significant impact on performance depending on the size of batch updates.

1. **Re-weight**: As seen in Eq. 7, we compute each particle's support on the new data batch:

$$u_i^{(r)} = \left[ \prod_j^{N_r} q_{\phi_i}\left(\theta_j^{(r)}|x_j^{(r)}\right) \right]^{\beta_r} \quad ; \quad w_i^{(r)} = \frac{u_i^{(r)}}{\sum_{k=1}^{K} u_k^{(r)}} \tag{43}$$

where $\phi_1, \ldots, \phi_K$ are assumed to initially make up an *unweighted* sample from $p(\phi|D)$, and $\beta_r$ is a round-wise annealing schedule $\{0, \ldots, 1\}$ (smooths sharp weight updates for large $N_r$). The effective model weights under the newly updated posterior can then be seen as

$$p(\phi_i|D^{(r+1)}) = \frac{p(\theta'|x', \phi_i)p(\phi_k|D^{(r)})}{p(\theta'|x', D^{(r)})} \tag{44}$$

$$= w_i^{(r)} \cdot p(\phi_i|D^{(r)}) \tag{45}$$

2. **Re-sample**: Draw an *unweighted* particle group by resampling $\hat{\phi}_i \sim$ Categorical $\left(w_1^{(r)}, \ldots, w_K^{(r)}\right)$.

3. **Re-train**: Run $S$ SGD steps to rejuvenate and diversify each model under a weighted likelihood bootstrap (WLB) scheme: $\nu_j \sim \text{Dirichlet}(1, \ldots, 1)$ for each $(\theta_j, x_j) \in D^{(r)}$.

Note that this procedure can be run several times of the dataset collected at the current $D^{(r)}$ to approximate increasingly small, refined steps through the model space $\Phi$. A balanced practical scheme should identify a trade-off between the frequency of chunked updates each round and computational budget; frequent re-weighting and re-training can be expensive, for instance, but generally facilitates better preservation of model diversity between sampling stages.

# F  ADDITIONAL ANALYSES

## F.1  TRADEOFFS WITH BATCHED METHODS

Non-active ESNPE can occasionally outperform the active mutual information variants, as can be seen in Table 1. While each method is unbiased and will converge to the true posterior in the limit as $N \to \infty$, behavior over small sample sizes in data scarce settings is subject to large amounts of variation. Ensembling in general can help improve consistency and robustness in these settings (compared to single SNPE-C runs, for instance), and naturally features a form of active selection when refining the posterior estimate round after round during inference. Ideally, the MI selection procedure helps capitalize on model uncertainty to improve parameter guidance, but estimating

model uncertainty is itself subject to noise and can at times lead the model astray. Across our experiments, we find the full batch MI variant is the single best go-to method, but in practice one should not expect a one-size-fits-all approach or realistically expect the active methods to uniformly outperform non-active ESNPE. We hypothesize that the active MI approaches are more likely to exhibit a performance advantage when there is early disagreement across component models under the prior. For instance, in Figure 3B, one can see explicit variability across component models as the ensemble attempts to find a consistent explanation for the first few rounds of simulations. This clear "disagreement" across models was more apparent on the Two Moons setting in our experiments, which may go some way to explaining the consistent batch MI outperformance on that environment in particular.

## F.2 LIMITATIONS OF EXISTING UNCERTAINTY-DRIVEN SBI METHODS

In Section 3.1, we referenced four related works (Järvenpää et al., 2019; Lueckmann et al., 2019; Griesemer et al., 2024; Krouglova et al., 2025) that leverage the posterior over model weights, in one form or another, to capitalize on uncertainty in the density estimator. Each approach has express limitations that make it incompatible or non-trivial to adapt to the APT setting, which in part motivates the flexibility in our formulation. In short, each existing method either cannot operate over arbitrary proposal distributions or involves optimizations that bias the posterior estimate:

1. In Järvenpää et al. (2019), the authors recognize explicitly that their use of an acquisition function (Eq. 2 in the paper) can bias the proposal distribution, "causing the density estimate to diverge from the true posterior." They further acknowledge that leveraging atomic proposals could be one way to correct for this, but avoid embracing any such corrective factor due to leakage concerns and complications adapting their proposed loss function.

2. In Lueckmann et al. (2019), the proposed approach leverages an ensemble of likelihood surrogates, not posterior models. This entails an entirely different set of tradeoffs, and cannot be directly used in the APT scheme. Further, this approach shares similar constraints to Järvenpää et al. (2019) in that parameters are selected via explicit optimization of an acquisition function (due to not upholding prior support, for instance), and is not compatible with arbitrary proposal distributions (making its adaptation to sequential settings non-trivial).

3. In Krouglova et al. (2025), the authors present several means of quantifying uncertainty in the posterior estimate, but they rely entirely on the use of Gaussian processes (GPs) for density estimation. This severely limits the portability of these techniques to new classes of generative models, e.g., flow-based models, and like Järvenpää et al. (2019) leverage optimization of an acquisition function that may violate the prior support across multi-round inference.

4. In Griesemer et al. (2024), a BNN / MC-dropout approach is used to implicitly model uncertainty in the posterior estimate. This is formulated as a variation of SNPE, but like Järvenpää et al. (2019), can bias the posterior estimate due to "blind spots" where the acquisition step effectively assigns zero likelihood during parameter selection.

## F.3 COMPUTATIONAL COMPLEXITY AND EMPIRICAL RUNTIMES

The non-active ESNPE ensembling scheme has an increased training time burden linear in $K$ compared to SNPE-C. Let $T_N$ be the time it takes to train a neural density estimator (NDE) over $N$ samples for one round of SNPE-C. If we carry out inference across $R$ rounds, the training time complexity of SNPE-C is $O(RT_N)$. Non-active ESNPE increases this to $O(KRT_N)$, accounting for the $K$ models that will each be trained on all $N$ samples each round. Note that the inference pipeline has a bottleneck during simulation: we must train the NDE before drawing proposal parameters for simulation, but we only simulate once per round. We can therefore parallelize the training of the $K$ ensemble models over the previous round's data, easing the additional training burden. In Table 3, we report wallclock times for full ESNPE inference runs with values of $K = 4, 8, 16$ with and without the batch MI selection step, as well as reference times under SNPE-C and T-SNPE. Reported times are averaged over five runs on the three benchmark environments from Table 1.

Table 3: Wallclock runtimes for batched and non-batched methods of ESNPE at various values of $K$, along with baseline runtimes of SNPE-C and TSNPE routines.

| Simulator | Method | Metric | |
| --- | --- | --- | --- |
| | | Wallclock time (s) | Wallclock time, batched (s) |
| Two moons | SNPE-C | $17.3 \pm 1.2$ | – |
| | T-SNPE | $34.9 \pm 1.5$ | – |
| | ESNPE ($K = 4$) | $30.1 \pm 1.1$ | $30.3 \pm 2.8$ |
| | ESNPE ($K = 8$) | $61.1 \pm 6.1$ | $65.1 \pm 0.5$ |
| | ESNPE ($K = 16$) | $118.2 \pm 13.4$ | $126.4 \pm 9.3$ |
| Gaussian mixture | SNPE-C | $15.0 \pm 2.1$ | – |
| | T-SNPE | $34.5 \pm 2.7$ | – |
| | ESNPE ($K = 4$) | $25.2 \pm 1.3$ | $26.1 \pm 1.5$ |
| | ESNPE ($K = 8$) | $46.4 \pm 2.3$ | $47.99 \pm 1.4$ |
| | ESNPE ($K = 16$) | $101.1 \pm 15.5$ | $102.2 \pm 2.4$ |
| Bernoulli GLM | SNPE-C | $26.1 \pm 3.7$ | – |
| | T-SNPE | $103.0 \pm 3.2$ | – |
| | ESNPE ($K = 4$) | $44.8 \pm 1.5$ | $48.5 \pm 1.5$ |
| | ESNPE ($K = 8$) | $108.7 \pm 2.7$ | $118.2 \pm 3.5$ |
| | ESNPE ($K = 16$) | $304.9 \pm 6.3$ | $307.2 \pm 27.1$ |

Here we find that parallel processing brings the ESNPE training time multiplier on $T_N$ to roughly $K/2$ (on our hardware): across each setting, we see that the $K = 4$ incurs approximately double the wallclock time of the single-model SNPE-C reference, for instance. Training ESNPE with $K = 4$ is often faster than single-model T-SNPE (which incurs additional time due to sampling), demonstrating parity with commonly used SBI methods with a small but feasible setting of $K$. Recall that all experiments reported in the manuscript ran ESNPE with $K = 8$.

Additionally note the marginal extra time burden of batched MI ESNPE. While the full batch MI selection procedure has a time complexity of $O(BMNK)$ (see Appendix A.2 for full details), it consists almost entirely of fast matrix multiplication on the GPU and employs dynamic programming to minimize wasted computation across the $B$ selected points. Above we find that the additional time incurred for the selection procedure is more or less negligible at this scale ($B = 256$, $N = 1024$, $M = 256$, $K = \{4, 8, 16\}$).

# G  ADDITIONAL RESULTS

## G.1  TASK DIMENSIONALITY

Table 4 captures a brief summary of the parameter and output dimensionality for each of the tasks/environments reported in the paper.

Table 4: Task $\theta$ and $x$ dimensions.

| Task | $\theta$-dim | $x$-dim |
| --- | --- | --- |
| Two moons | 2 | 2 |
| Gaussian mixture | 2 | 2 |
| Bernoulli GLM | 10 | 10 |
| SIR | 2 | 10 |
| Lotka Volterra | 4 | 20 |

Table 5: Results comparing ESNPE variants with baseline methods for C2ST (means and 95% CIs over 10 trials) on the *Bernoulli GLM*, *Gaussian mixture*, and *Two moons* tasks. Metrics include both C2ST and MMD, and each trial averages over all 10 reference posterior slices $p(\theta|x_1), \ldots, p(\theta|x_{10})$.

| *Simulator* | *Method* | *Metric* | |
| | | C2ST | MMD |
|---|---|---|---|
| *Two moons* | SMC-ABC | $0.986 \pm 0.003$ | $0.068 \pm 0.102$ |
| | FMPE | $0.966 \pm 0.036$ | $0.194 \pm 0.095$ |
| | NPSE | $0.890 \pm 0.045$ | $0.017 \pm 0.026$ |
| | TSNPE | $0.936 \pm 0.015$ | $0.069 \pm 0.053$ |
| | SNPE-C | $0.843 \pm 0.037$ | $0.018 \pm 0.011$ |
| | ESNPE | $\mathbf{0.811 \pm 0.025}$ | $\mathbf{0.014 \pm 0.005}$ |
| | ESNPE-B-MI | $0.828 \pm 0.030$ | $0.019 \pm 0.014$ |
| *Gaussian mixture* | SMC-ABC | $0.888 \pm 0.012$ | $0.357 \pm 0.081$ |
| | FMPE | $0.884 \pm 0.015$ | $0.368 \pm 0.057$ |
| | NPSE | $0.737 \pm 0.028$ | $0.120 \pm 0.068$ |
| | TSNPE | $0.779 \pm 0.016$ | $0.189 \pm 0.062$ |
| | SNPE-C | $0.717 \pm 0.012$ | $0.023 \pm 0.013$ |
| | ESNPE | $\mathbf{0.716 \pm 0.010}$ | $0.022 \pm 0.012$ |
| | ESNPE-B-MI | $\mathbf{0.716 \pm 0.015}$ | $\mathbf{0.020 \pm 0.008}$ |
| *Bernoulli GLM* | SMC-ABC | $0.991 \pm 0.002$ | $0.256 \pm 0.077$ |
| | FMPE | $0.956 \pm 0.021$ | $0.199 \pm 0.045$ |
| | NPSE | $0.901 \pm 0.042$ | $0.215 \pm 0.157$ |
| | TSNPE | $0.952 \pm 0.013$ | $0.230 \pm 0.125$ |
| | SNPE-C | $0.760 \pm 0.066$ | $0.124 \pm 0.108$ |
| | ESNPE | $0.739 \pm 0.048$ | $0.098 \pm 0.098$ |
| | ESNPE-B-MI | $\mathbf{0.733 \pm 0.070}$ | $\mathbf{0.064 \pm 0.042}$ |

## G.2 BENCHMARKING ALL REFERENCE POSTERIORS

The three individual observations originally provided in Table 1 were selected and reported independently to highlight method performance across variable reference data, as opposed to averaging out performance across all available $x_o$ samples (with differences in posterior shape visualized in Figure 3, for instance). To provide a more holistic analysis, we ran the same experiments from Table 1 across all available reference observations in Lueckmann et al. (2021) for the three primary tasks, and report both the average C2ST and MMD scores in Table 5. Note that we only re-evaluated the non-active and full batch MI ESNPE variants for ease of comparison.

Here we find our method exhibits a performance advantage across all settings similar to that observed on the three individual observations reported in the paper, and the MMD metric tends to agree with C2ST in terms of the best scoring approach.

## G.3 HYPERPARAMETER ABLATIONS

We carried out several ablations across hyperparameters $K$ and $B$ to shed light on how the method performs under variable conditions. When holding the number of simulation samples constant ($N = 256$ across $R = 4$ rounds, just as reported in Table 1), we find a clear trend in performance as we increase the number of component models $K$ in Table 6.

The values for each metric are means with standard deviations calculated across five runs. Here we more or less observe the expected relationship between ensemble size and performance: the more component models we have, the more stable the posterior approximation round-by-round, and these compounding often translate to increased posterior quality. Note that "ESNPE" here refers to the non-active variant of the method.

Table 6: Ablations for non-batched ESNPE under variable settings of $K$.

| | | Metric | |
|---|---|---|---|
| *Simulator* | *Hyperparam* | C2ST | MMD |
| *Two moons* | $K = 1$ | $0.961 \pm 0.005$ | $0.064 \pm 0.029$ |
| | $K = 4$ | $0.954 \pm 0.002$ | $0.064 \pm 0.003$ |
| | $K = 8$ | $0.929 \pm 0.025$ | $0.049 \pm 0.004$ |
| | $K = 16$ | $0.928 \pm 0.025$ | $0.039 \pm 0.009$ |
| *Gaussian mixture* | $K = 1$ | $0.901 \pm 0.006$ | $0.655 \pm 0.042$ |
| | $K = 4$ | $0.753 \pm 0.009$ | $0.165 \pm 0.019$ |
| | $K = 8$ | $0.744 \pm 0.006$ | $0.024 \pm 0.013$ |
| | $K = 16$ | $0.736 \pm 0.021$ | $0.014 \pm 0.006$ |
| *Bernoulli GLM* | $K = 1$ | $0.999 \pm 0.001$ | $0.738 \pm 0.045$ |
| | $K = 4$ | $0.992 \pm 0.008$ | $0.803 \pm 0.271$ |
| | $K = 8$ | $0.956 \pm 0.009$ | $0.480 \pm 0.030$ |
| | $K = 16$ | $0.933 \pm 0.017$ | $0.321 \pm 0.032$ |

Table 7: Ablations for batched ESNPE under different numbers of inference rounds $R$.

| | | Metric | |
|---|---|---|---|
| *Simulator* | *Hyperparam* | C2ST | MMD |
| *Two moons* | $R = 2$ | $0.805 \pm 0.002$ | $0.011 \pm 0.001$ |
| | $R = 4$ | $0.856 \pm 0.014$ | $0.014 \pm 0.002$ |
| | $R = 8$ | $0.925 \pm 0.002$ | $0.025 \pm 0.009$ |
| *Gaussian mixture* | $R = 2$ | $0.692 \pm 0.016$ | $0.018 \pm 0.006$ |
| | $R = 4$ | $0.707 \pm 0.012$ | $0.022 \pm 0.010$ |
| | $R = 8$ | $0.713 \pm 0.022$ | $0.026 \pm 0.002$ |
| *Bernoulli GLM* | $R = 2$ | $0.769 \pm 0.056$ | $0.182 \pm 0.058$ |
| | $R = 4$ | $0.811 \pm 0.019$ | $0.216 \pm 0.050$ |
| | $R = 8$ | $0.823 \pm 0.003$ | $0.230 \pm 0.007$ |

We additionally run ablations over the number of samples drawn per round, or the batch size of the active MI ESNPE. Values are reported in Table 7. As before, we hold the total number of simulation samples $N = 1024$ fixed across the entire inference run with $K = 8$, but vary the number of round $R$ over which those samples are collected. Smaller values of $R$ therefore correspond to larger batch sizes for the MI selection mechanism.

We observe a fairly clear trend here as well, but perhaps counter-intuitively find that having fewer rounds reliably leads to better performance. Generally, we might expect more rounds to provide more opportunity for the proposal distribution to be updated to the current posterior approximation. However, given the relatively small value of $N$, smaller $R$ translates to more stable collections of samples from the simulator across rounds. For $R = 8$, each round provides just 128 samples, often yielding noisy updates in the density estimators that counteract the active selection mechanism.

## H   ADDITIONAL EXPERIMENTAL DETAILS

The NDE model used for all experiments reported in Table 1 is a conditional masked autoregressive flow (MAF) comprised of 5 MADE transforms, each with two 50-unit residual blocks. For experiments in Table 2, MAFs with 5 transformers and 128-unit residual blocks were used. Each MAF was trained with 10 atoms, and we use most of the training core as implemented in the sbi Python package Tejero-Cantero et al. (2020). In Table 1, sequential methods were run across four inference rounds with 256 samples per round, for a total of 1024 training examples. In Table 2, 256

samples per round was kept fixed, but inference runs were carried out across eight rounds, for a total of 2048 training examples. No embedding networks were employed across any of the tasks, and the `sbibm` package Lueckmann et al. (2021) was adapted to provide true posterior references and metric calculations for each reported task.

All experimentation code was written and packaged in Python 3.13. Experiments were performed on local hardware: an Arch linux-based machine running an Intel(R) Core(TM) i9-10900X CPU @ 3.70GHz 64GB memory and NVIDIA GeForce RTX 2080 Ti. All experiments, baseline models, and benchmark environments are reproducible from a centralized Python package, to be released upon publication.

### H.1 REPRODUCIBILITY DETAILS

All reported baseline methods use default implementation and hyperparameters, unless otherwise specified, as set in the open source `sbi` Python package Tejero-Cantero et al. (2020), allowing for independently verifiable results rooted in out-of-the-box standards. The pinned version of `sbi` used for all experiments is the stable `0.24.0` release. Although all tasks/environments for our experiments were adapted directly from the data reported in Lueckmann et al. (2021), one can reproduce the exact numbers reported in Table 5, for instance, with the convenient `sbibm-mini` module that ships with the `sbi` package. Below is a minimal sketch for quickly verifying these results:

1. Download the source of the `v0.24.0` stable release of the `sbi` package from Github (`v0.24.0` was used for all experiments) and install locally (with `[dev]` extras) in an isolated Python 3.11 environment.

2. Under the `sbi` package root in `tests/bm_test.py`:
   (a) on line 20, set: `NUM_SIMULATIONS = 1024`
   (b) on line 21, set: `NUM_EVALUATION_OBS = 10`
   (c) on line 22, set: `NUM_ROUNDS_SEQUENTIAL = 4`

3. Run `pytest` for each benchmark mode, yielding amortized evaluation over all 10 observations, e.g., `pytest –bm –bm-mode=fmpe`.

4. Run the same `pytest` commands above for 10 seeds (manually setting the `SEED` variable on line 18 to values $1, \ldots, 9$; the default `SEED=0` assumed to be run in the above step).

Taking the mean and standard deviation across the 10 seeds should yield nearly identical results, for both C2ST and MMD across all methods, to those reported in Table 5. Do note that FMPE and NPSE are not sequential methods, and therefore produce an amortized posterior that is evaluated under all 10 reference observations. SNPE-C, on the other hand, is used as a non-amortized method, conditioning on a single reference observation $x_o^{(i)}$ across inference rounds, and the resulting posterior is evaluated only under $x_o^{(i)}$. The `sbibm-mini` module as is requires changing the `NUM_EVALUATION_OBS_SEQ` variable across individual benchmark runs to recover non-amortized evaluation over all 10 observations.

