# OpenReview forum: "Informative Posterior Ensembles for Sequential Simulation-Based Inference"
_ICLR.cc/2026/Conference — Submitted to ICLR 2026_

### Official Review · Reviewer_p85r · 2025-10-31

**Soundness:** 3
**Presentation:** 3
**Contribution:** 3
**Rating:** 4
**Confidence:** 4

**Summary:**

The paper introduces ESNPE (Ensemble Sequential Neural Posterior Estimation), with the overall goal of improving sample-effiency in sequential simulation-based inference through:
- Posterior ensembling (and introducing a method that does this coherently)
- A mutual information-based candidate parameter point acquisition scheme, which grabs parameters that on average reduce model uncertainty
- A batch acquisition algorithm (jointly select informative batches)

**Strengths:**

- Overall, rather than a single contribution this paper is a collection of technical extensions that are aimed at improving sample efficiency and information gain during SBI.
- The ensembling makes sense and is theoretically guaranteed. Overall, compatibility is maintained with a previous approach APT's atomic proposal framework, giving asymptotic correctness
- Mutual information as a parameter acquisition metric is solid and a nice contribution

**Weaknesses:**

- The residual MI method seems to assume every candidate parameter \theta' will yield the observation. \theta' ~ p(x|\theta') produces different outputs in practice, which seems like it might violate this assumption?
- Since the authors introduce a few different tweaks to SNPE, it seems important to understand which parts are contributing to the performance (e.g, MI-based acquisition, ensembling, etc)
- From the results, it seems that active variants sometimes underperform compared to non-active ones, leading to the overall utility being a bit questionable (or limited to specific cases, which could be fleshed out for practitioners to understand when the method can help). It looks like fewer rounds can often be helpful in experiments, which seems surprising and maybe undermines the basic sequential aspect of the algorithm?
- The applications are fairly simple (e.g. typical benchmarks, low-dimensional) -- it seems typical these days to ground SBI expositions with more realistic / real-world use cases

**Questions:**

Questions are primarily based on weaknesses above:
- Is the "residual" MI approximation actually valid as laid out? Could it be checked with toy models where ground truth is known?
- What are the key performance drivers -- e.g., MI-based parameter selection, ensembling, etc?
- Why do active variants sometimes underperform (could it be the noisy MI estimates)?

---

> ### Author Response · Authors · 2025-11-23
> **Rebuttal (1/2)**
>
> Thank you for your time and effort providing detailed feedback on our work. Please find our responses to your questions and comments below.
>
> **Note**: Where applicable, we prefix sections with `W-<x>` or `Q-<y>`to reference itemized comments in Weaknesses and Questions, respectively, numbered by the order in which they were mentioned. Related items are grouped accordingly.
>
> ---
>
> **[W-1]** Candidate parameters are not assumed to directly produce $x_o$; instead, $\theta_i$’s are explicitly paired with $x_o$ when conditioning the mutual information term (e.g., as seen in line 242), and we then optimize for the best candidate $\theta_i$ under this assumption. This is less of an exact constraint being imposed and more of a way to reason about which parameters are most likely to yield the outcome we’re seeking, even if the pairing $(\theta_i,x_o)$ is explicit. This is loosely what’s meant by “residual mutual information,” and the mechanism by which we successively build up parameter batches expected to be informative under the data $x_o$.
>
> **[W-2], [Q-2]** We attempt to isolate the contribution of each proposed strategy by directly evaluating variants of the base ensembling method (ESNPE). In Table 1, for instance, ESNPE, the naive active extension (ESNPE-MI), and the batched acquisition scheme (ESNPE-B-MI) are all reported separately, highlighting the role of parameter selection in the sequential inference process. There are several caveats that make it difficult to pin a single best method (see Appendix F1 for an in-depth discussion), but results shown in primary tables serve as ablations over the proposed additions (alongside the more pointed ablations in the Appendix, e.g., Tables 5, 6, 7).
>
> *To include a more qualitative assessment*: the ensembling and batch selection work in tandem, contributing more or less to a performance advantage in different environments. The ensembling scheme facilitates the inclusion of more diverse posterior representations, and batch acquisition attempts to capitalize on uncertainty across these posterior realizations. In practice, one finds that there is noise across these strategies that can affect the quality of approximation, but as seen across Tables 1/2/5, they reliably offer performance benefits over methods lacking these components. Table 5 in particular highlights how batch selection can provide useful signal in some instances (as in Bernoulli GLM) while appearing to add noise in others (as in Two Moons), even when outperforming baselines.
>
> **[W-3], [Q-3]** This is a great point. Systematically qualifying precisely when one version of the method (active or not) will perform better on a given environment is challenging due to variable performance across different environments. In practice, as is the case for selection between existing SBI methods, this is difficult to anticipate without some initial brute force evaluation on a specific task. As alluded to above, Appendix F1 attempts to characterize when batched selection may encounter practical limitations (e.g., due to noisy approximations) and goes some way toward practical guidance when attempting to square the tradeoffs. In any case, we believe experiments on larger simulation budgets and additional environments are likely to better reveal possible tradeoffs across method variants, both of which are currently being undertaken; please the new experiment details below.
>
> *On numbers of rounds*: The behavior under increasing numbers of rounds reported in Appendix G3 is more indicative of dynamics at this specific simulation budget rather than a method limitation. As mentioned in the corresponding analysis, this is downstream of increasingly noisy round-wise updates, and an identical trend is observed for vanilla SNPE-C at this simulation budget. For larger budgets, one can expect to see more rounds lead to better metrics (up to a point), as the *absolute size* of the batch collected each round is important for model stability.
>
> **[W-4]** We agree that many of the core SBIBM tasks are toy-like in nature. However, these environments are prioritized due to their ease in reproducibility across the field, and are critically coupled with true posterior reference samples, allowing for concrete quantitative analyses (e.g., with C2ST and MMD metrics). Additionally, under limited sample budgets, we believe these environments do present a reasonable challenge for SBI methods, as is apparent in our reported C2ST values. For instance, with the Two Moons task at the ~1k sample mark, most baseline methods are unable to breach the 0.9 C2ST mark, indicating a sizable gap between the approximate and true posterior. We are nevertheless actively running experiments on two higher dimensional settings; please see the details on new experiments below.

---

> ### Author Response · Authors · 2025-11-23
> **Rebuttal (2/2)**
>
> **[Q-1]** The residual MI term is downstream of a standard mutual information construction, as derived in Appendix A1. Ground truth values could indeed be calculated on a simple toy model, as these would be identical to the direct calculations made by the non-batched selection mechanism (i.e., the MI terms for individual parameters are exact). The batched case is more difficult, however, due the stochastic selection of parameters. By design, our method never attempts to produce a fully optimal batch (we *must* randomly sample batch components to preserve posterior convergence guarantees), and directly comparing to a ground truth optimal batch would therefore not be particularly meaningful.
>
> ---
>
> ### **Summary of new experimentation**
> *(this takes quite some time on our hardware; we appreciate your patience as we collect results to update our manuscript)*
>
> - We are running existing experiments at larger simulation budgets (~10x current max sample size) and are recording round-wise results in order to produce plots rather than purely tabular, final-round metrics.
> - We’re including two higher dimensional real-world environments to shed light on outstanding scalability concerns: the stomatogastric ganglion pyloric network [1] (31-dim $\theta \rightarrow$ 18-dim $x$) and the Hodgkin-Huxley model [2] (7-dim $\theta \rightarrow$ large noisy time series). Although these environments lack reference posterior samples for clean posterior quality evaluations (such as C2ST), they are widely analyzed across SBI literature (e.g., in [3], [4]) and a qualitative precedent exists when evaluating posterior predictive coverage.
>
> We’re more than happy to address any remaining concerns or provide additional clarification where needed.
>
> ---
>
> **References**
> **[1]** *Deistler, Michael, Jakob H. Macke, and Pedro J. Gonçalves. "Energy-efficient network activity from disparate circuit parameters." Proceedings of the National Academy of Sciences 119.44 (2022): e2207632119.*
> **[2]** *Gonçalves, Pedro J., et al. "Training deep neural density estimators to identify mechanistic models of neural dynamics." elife 9 (2020): e56261.*
> **[3]** *Deistler, Michael, Pedro J. Goncalves, and Jakob H. Macke. "Truncated proposals for scalable and hassle-free simulation-based inference." Advances in neural information processing systems 35 (2022): 23135-23149.*
> **[4]** *Gloeckler, Manuel, et al. "All-in-one simulation-based inference." arXiv preprint arXiv:2404.09636 (2024).*

---

### Official Review · Reviewer_dTBE · 2025-10-31

**Soundness:** 3
**Presentation:** 3
**Contribution:** 2
**Rating:** 4
**Confidence:** 4

**Summary:**

This paper proposes a novel method for sequentiel neural posterior estimation, optimizing for the proposed parameter values to improve information gain for the target posteiror p(theta | x_0). I am short on time due to the semester start. Apologies if my reviews are a bit short. I am happy to engage in reviewer discussion should be concerns not be clear. And I am happy to consider increasing my score should the authors provide convincing responses to my concerns.

**Strengths:**

- The new method is theoretically motivated and appears overall sound. Common issues arising in SNPE are tackled and avoid in sensible ways.
- The writing of the paper is clear, at least to people familiar with the literature already.
- The new method beats alternative methods in common benchmarks, although the benchmarks themselves are too simple (see below).

**Weaknesses:**

- The benchmarks used in the paper are very simply models, which continue to be reused in papers but do not reflect actually challenging SBI tasks. I don't mind using some of them as toy examples. But having no actually challenging models in the paper is insufficient in my opinion. Neither SIR nor LV are challenging models for NPE in general (nor the other 3 even more toy examples). In particular, the authors claim that their method is particularily useful for expensive simulators which only allow for very small simulation budgets, but non of the evaluated models has an expensive simulator (quite the opposite in fact). I would have liked to see at least one challenging case study with a complex, expensive simulator that really shows the benefits of the method.

- Two amortized methods (FMPE and NPSE) are considered and tend to imply as good results as many of the sequential methods (an observation also made in other papers; e.g., https://arxiv.org/abs/2302.09125, not cited by the authors) — sometimes even reaching your ESNPE method in the accuracy metrics. Yet, the fact that we can reach almost as good accuracy in many cases while obtaining amortized posteriors across the whole prior space, is not addressed or discussed properly in this paper.

- While, in the accuracy metrics, amortized methods are included, they do not show up in the speed comparisons? Why? I think a fair comparison would demand them to appear also there. I may have overlooked these results but at least Table 3 in the appendix didn't provide them.

- The literature is cited very selectively, largely citing papers from few selective groups, an issue I continue to see in the SBI literature. This gives newcomers to the field a wrong sense of what is done by whom and leads to a fragmentation of the literature. As an example, the works of Stefan Radev are not cited even once, although he has made many relevant contributions to the field. I am seeing this selective referencing in almost every major ML conference, favoring some groups over others (not always the same groups) and it increasingly irritates me. Please cite more diverse works in your papers, not just papers from (presumably) your own group and few adjacent groups.

**Questions:**

- How does the method scale with parameter dimensionality? The considered benchmarks are all very low dimensional and I wonder if there is any potential scalability issue of the new method?
- What do you mean with (1 - 1/e - epsilon)-approximate concretely?
- How are coupling flows performing for these examples? Likely not good for the multimodal models but for the others, the would provide a good baseline, with very high inference time speed.

---

> ### Author Response · Authors · 2025-11-23
> **Rebuttal (1/2)**
>
> Thank you for your time and effort providing detailed feedback on our work. Please find our responses to your questions and comments below.
>
> **Note**: Where applicable, we prefix sections with `W-<x>` or `Q-<y>`to reference itemized comments in Weaknesses and Questions, respectively, numbered by the order in which they were mentioned. Related items are grouped accordingly.
>
> ---
>
> **[W-1], [Q-1]** We agree that many of the core SBIBM tasks are toy-like in nature. However, these environments are prioritized due to their ease in reproducibility across the field, and are critically coupled with true posterior reference samples, allowing for concrete quantitative analyses (e.g., with C2ST and MMD metrics). Additionally, under limited sample budgets, we believe these environments do present a reasonable challenge for SBI methods, as is apparent in our reported C2ST values. For instance, with the Two Moons task at the ~1k sample budget, most baseline methods are unable to breach the 0.9 C2ST mark, indicating a sizable gap between the approximate and true posterior.
>
> *Regarding scalability*: Given the underlying reliance on SNPE-C, the core ensembling technique can be expected to scale similarly to this method. To better characterize this, and address the concern from [W-1], we are actively running experiments on two higher dimensional settings; please see the details on new experiments below.
>
> **[W-2]** The primary focus of the paper is indeed non-amortized methods, and centers around formulating an ensembling scheme (plus uncertainty-driven parameter exploration) across multiple rounds of sequential inference. As such, while amortized methods like FMPE and NPSE are included as baselines, carrying out a full analysis of tradeoffs between amortized and non-amortized approaches is not strictly within the scope of the paper. Amortized methods no doubt offer important benefits when one cares about posterior quality beyond a single slice at $x_o$, and there are many studies that indicate as much. However, focusing directly on the task of learning an accurate $p(\theta|x_o)$ under a fixed simulation budget allows for more targeted experimentation when evaluating sequential methods, and we therefore focus exclusively on this task formulation. Nevertheless, a wider discussion that ensures readers and practitioners are aware of the broad tradeoffs between amortized and non-amortized methods is warranted, and a corresponding discussion will be included in the updated manuscript (for which the mentioned reference *Radev et al., 2023* provides useful context).
>
> **[W-3]** The primary goal of Table 3 is to highlight the relationship between training time and increasingly large ensemble sizes rather than fully characterizing runtime tradeoffs for all reported methods (hence the omission of ASNPE, SMCABC, as well as the amortized methods). SNPE-C and TSNPE were reported as the most directly comparable in terms of procedural similarity, as they employ the same underlying NDE model. We do, however, believe that a full accounting of runtime tradeoffs across reported baseline methods would be an insightful addition, and will endeavor to include this in our updated manuscript.
>
> **[W-4]** Thank you for calling attention to this. We should note the literature review was not intentionally selective for specific groups but instead picked up on more widely embraced SBI procedures, especially those with out-of-the-box implementations within or adjacent to the `sbi` package. We nevertheless agree this is likely to reflect common citation patterns, and works from Stefan Radev, among others, are worth exploring in greater detail given the overlap. Corresponding references and discussion will be added in our updated manuscript.
>
> **[Q-2]** The precise meaning behind the $(1 - 1/e - \epsilon)$-approximate scheme is described in Appendix C2.2, which is downstream of the generic approximate algorithm construction as presented in Nemhauser et al., 1978. $(1 - 1/e - \epsilon)$ is the factor by which the stochastic, greedily collected parameter batch underperforms the optimal batch with respect to the mutual information objective $\mathbb{M}(\Theta) = \mathbb{I}[\phi;\theta|x_o, D\cup\{(\theta^\prime, x_o)\}_{\theta^\prime\in\Theta}]$, which is shown to be submodular in Lemma C2. The $1/e$ concession results from greedily maximizing this objective one parameter at a time, and the $\epsilon$ factor results from selecting these parameters at random (so long as in expectation we make no greater than a $\epsilon$ concession from the next best parameter; see Lemma C3).

---

> ### Author Response · Authors · 2025-11-23
> **Rebuttal (2/2)**
>
> **[Q-3]** Coupling flows were not directly used in our experiments as we kept the density estimator for SNPE-adjacent methods fixed to masked autoregressive flows. This choice prioritizes expressiveness while aligning reasonably well with our method’s sample/likelihood evaluation requirements (i.e., few sample draws, many model likelihood evaluations across ensemble components). Coupling flows would be an interesting alternative to explore in follow-up experiments where a model with both fast sampling/likelihood evaluation is preferred, if less expressive.
>
> ---
>
> ### **Summary of new experimentation**
> *(this takes quite some time on our hardware; we appreciate your patience as we collect results to update our manuscript)*
>
> - We are running existing experiments at larger simulation budgets (~10x current max sample size) and are recording round-wise results in order to produce plots rather than purely tabular, final-round metrics.
> - We’re including two higher dimensional real-world environments to shed light on outstanding scalability concerns: the stomatogastric ganglion pyloric network [1] (31-dim $\theta \rightarrow$ 18-dim $x$) and the Hodgkin-Huxley model [2] (7-dim $\theta \rightarrow$ large noisy time series). Although these environments lack reference posterior samples for clean posterior quality evaluations (such as C2ST), they are widely analyzed across SBI literature (e.g., in [3], [4]) and a qualitative precedent exists when evaluating posterior predictive coverage.
>
> We’re more than happy to address any remaining concerns or provide additional clarification where needed.
>
> ---
> **References**
> **[1]** *Deistler, Michael, Jakob H. Macke, and Pedro J. Gonçalves. "Energy-efficient network activity from disparate circuit parameters." Proceedings of the National Academy of Sciences 119.44 (2022): e2207632119.*
> **[2]** *Gonçalves, Pedro J., et al. "Training deep neural density estimators to identify mechanistic models of neural dynamics." elife 9 (2020): e56261.*
> **[3]** *Deistler, Michael, Pedro J. Goncalves, and Jakob H. Macke. "Truncated proposals for scalable and hassle-free simulation-based inference." Advances in neural information processing systems 35 (2022): 23135-23149.*
> **[4]** *Gloeckler, Manuel, et al. "All-in-one simulation-based inference." arXiv preprint arXiv:2404.09636 (2024).*

---

> > ### Comment · Reviewer_dTBE · 2025-11-26
> >
> > Thank you for your responses. I decided to keep my score for now but remain open to change my score also depending on the perspectives of the other reviewers.

---

### Official Review · Reviewer_GiCU · 2025-11-03

**Soundness:** 2
**Presentation:** 2
**Contribution:** 2
**Rating:** 2
**Confidence:** 4

**Summary:**

The paper describes an active learning method for neural posterior estimation (NPE). They propose a (weighted)) ensemble of the neural density estimator and provide an active learning rule based on the mutual information of parameters and neural network weights. They also provide a batched version of this rule. They evaluate their method on popular SBI benchmark tasks.

**Strengths:**

The active learning rule presented here is novel (for NPE) and is theoretically grounded. The paper makes multiple advances for active learning in NPE. The figures are of high quality and aid in understanding the paper. Finally, the experiments are evaluated against a wide range of other methods.

**Weaknesses:**

My main issue is with the experimental results:

- the paper only evaluates the method on three (sometimes even two) observations. The used SBI benchmark provides ten observations, and it is unclear why these are not used.
- there is basically no difference in the performance between `ESNPE` and `ENSPE (batch MI)`. For example, on Bernoulli GLM, ESNPE outperforms batch MI, and on Gaussian mixture they perform exactly the same. Yet the authors write: "with the Batch MI acquisition generally yielding a larger performance advantage". It is unclear to me where this claim is coming from.
- Based on this observation, it seems to me that the only performance gain seems to come from ensembling. This, however, has been done before: For sequential NPE methods, for example, by Deistler et al., 2022. At the very least, this observation should be noted and highlighted in the paper.
- The performance of baseline methods is much poorer than what is reported in the respective original publications. For example, for Two moons, you report a C2ST of ~0.9 for SNPE-C or TSNPE. However, the TSNPE paper reports an average of ~0.65. Did you choose the exact same hyperparameters for SNPE-C (and TSNPE) as for ESNPE? For example, which density estimator did you use?

Beyond this, I find the appendix section F2 obscure:

- In paragraph 1 (Järvenpää), I could not find the quote "causing the density estimate to diverge from the true posterior." in the paper.
- In paragraph 2 (Lueckmann), it is unclear to me why it "is not compatible with arbitrary proposal distributions". It is a likelihood-based method and should, thus, but compatible with any kind of proposal.
- In paragraph 3 (Kourglova), you claim that they rely on Gaussian processes, but this is simply not true (and GPs are not even mentioned in the paper).

These errors are so bad that I am suspecting the paragraph to be written (unchecked) by a large-language model. I have therefore checked the corresponding LLM usage mark.

**Questions:**

I sometimes find the methods section hard to follow:

- is it correct that the ensemble weights are equal to exp(-loss)?
- Line 195: `by calculating its (normalized) support under`. Why does one have to compute a support?

---

> ### Author Response · Authors · 2025-11-23
> **Rebuttal (1/2)**
>
> Thank you for your insightful feedback on our work. Please find our responses to your questions and comments below.
>
> **Note**: Where applicable, we prefix sections with `W-<x>` or `Q-<y>`to reference itemized comments in Weaknesses and Questions, respectively, numbered by the order in which they were mentioned. Related items are grouped accordingly.
>
> ---
>
> **[W-1]** Table 5 in the Appendix provides results averaged over all ten available observations. The original intent in providing observation-specific results in Table 1 was to highlight variability across specific posterior slices, as mirrored in Figure 3 (which specifically visualizes Two Moons). This demonstrates nuance within individual settings rather than marginalizing over every instance. However, as indicated in our response to Reviewer `KpYD`, our updated manuscript will switch Table 1 and Table 5, highlighting the average results in the paper body while relegating the more specific analyses to the Appendix. We will additionally complement the final-round metrics provided Table 5 with full round-wise plots, making it easier to compare performance across the entire inference process.
>
> **[W-2]** The statement that the Batch MI variant holds a larger performance advantage is broad, but was directly downstream of the case-by-case results shown in Table 1. Among the nine settings, Batch MI was the best scoring method in six cases, outperforming the other variants for more than half of reported settings. This statement is not without caveat, however, and Appendix F1 attempts to characterize when batched selection may encounter practical limitations (e.g., due to noisy approximations).
>
> **[W-3]** Thank you for pointing to this reference, and the use of ensembling in Deistler et al., 2022 (i.e., TSNPE) is worth noting. However, in the corresponding paper [3], the discussion of ensembling methodology is very minimal, and the authors only employ ensembling within the “Multicompartment model of a single neuron” environment. In this case, the justification defers to observations as presented in Hermans et al. 2022 [5], but this work does not prescribe a specific means of systematically maintaining an ensembled posterior across multiple rounds of inference. We nevertheless believe this reference should be included in our discussion, and our updated manuscript will include these observations.
>
> **[W-4]** As mentioned in Appendix H, results for all methods are produced using out-of-the-box defaults from the `sbi` Python package [6]. For SNPE-based methods (SNPE-C, TSNPE, ENSPE), the default density estimator is a masked autoregressive flow (MAF), and this was the sole neural density estimator (NDE) tested across our experiments to better isolate the impact of the outer inference procedure. In the TSNPE paper [3], the authors primarily report results using neural spline flows, but the TSNPE method itself remains generic to the choice of NDE. Our experiments also differ in the total number of samples and rounds for inference; the results from the TSNPE are therefore not directly comparable. Appendix H1 details a simple procedure for directly reproducing reported results (e.g., Table 5) with out-of-the-box defaults from the `sbi` toolkit (which includes a TSNPE implementation directly from the original authors).
>
> **[W-5]** Thank you for calling attention to the citation error in Appendix F2. This paragraph was originally part of the main paper body, but was moved to the Appendix due to space constraints. In transition, it appears the "Kourglova" and "Järvenpää" references were accidentally permuted (i.e., the highlighted papers in bullet points 1 and 3). With these two references swapped back, the cited evidence in the respective bullet points lines up appropriately, i.e., Järvenpää et al. is a GP-centric paper while Kourglova et al. includes the mentioned quote from the first bullet. Reviewer `KpYD` also makes note of the accidental swap here, and we’d like to make clear this was merely a syntax error and *not* downstream of LLM-generated text. We reaffirm that LLMs were not used in the writing of this paper whatsoever.
>
> *Regarding Lueckmann et al. [7]*, our statement regarding flexibility around arbitrary proposals is lacking detail and confusingly worded in hindsight. In the general NLE sense, the likelihood model is indeed compatible with any proposal due to its independence of the prior. The proposed acquisition strategy (Eq. 1 in the paper), however, does depend directly on the true prior, and our statement was aimed at the non-trivial adjustments required to adapt these terms under other proposal distributions. This is exacerbated by the previously expressed issue under deterministic optimization of the acquisition objective, which can in principle systematically ignore regions of the parameter space that are within prior support. The wording in the manuscript will be revised to reflect the full nuance here.

---

> ### Author Response · Authors · 2025-11-23
> **Rebuttal (2/2)**
>
> **[Q-1]** Ensemble weights are calculated according to Eq. 7, i.e., the self-normalized importance weights under simulated samples. Assuming you’re referring to the $\exp(-\text{loss})$ form seen in Eq. 10, this is used to shape the *proposal* distribution over parameter candidates rather and is separate from the ensemble weight computation.
>
> **[Q-2]** The use of “support” in line 195 was meant in a more generic sense, as in how well new pairs $D^\prime \setminus D$ are evidentially supported by the model $\phi$. This term is fairly overloaded, however, making this sentence ambiguous; we will revise the wording here accordingly.
>
> ---
>
> ### **Summary of new experimentation**
> *(this takes quite some time on our hardware; we appreciate your patience as we collect results to update our manuscript)*
>
> - We are running existing experiments at larger simulation budgets (~10x current max sample size) and are recording round-wise results in order to produce plots rather than purely tabular, final-round metrics.
> - We’re including two higher dimensional real-world environments to shed light on outstanding scalability concerns: the stomatogastric ganglion pyloric network [1] (31-dim $\theta \rightarrow$ 18-dim $x$) and the Hodgkin-Huxley model [2] (7-dim $\theta \rightarrow$ large noisy time series). Although these environments lack reference posterior samples for clean posterior quality evaluations (such as C2ST), they are widely analyzed across SBI literature (e.g., in [3], [4]) and a qualitative precedent exists when evaluating posterior predictive coverage.
>
> We’re more than happy to address any remaining concerns or provide additional clarification where needed.
>
> ---
>
> **References**
> **[1]** *Deistler, Michael, Jakob H. Macke, and Pedro J. Gonçalves. "Energy-efficient network activity from disparate circuit parameters." Proceedings of the National Academy of Sciences 119.44 (2022): e2207632119.*
> **[2]** *Gonçalves, Pedro J., et al. "Training deep neural density estimators to identify mechanistic models of neural dynamics." elife 9 (2020): e56261.*
> **[3]** *Deistler, Michael, Pedro J. Goncalves, and Jakob H. Macke. "Truncated proposals for scalable and hassle-free simulation-based inference." Advances in neural information processing systems 35 (2022): 23135-23149.*
> **[4]** *Gloeckler, Manuel, et al. "All-in-one simulation-based inference." arXiv preprint arXiv:2404.09636 (2024).*
> **[5]** *Hermans, Joeri, et al. "A trust crisis in simulation-based inference? your posterior approximations can be unfaithful." arXiv preprint arXiv:2110.06581 (2021).*
> **[6]** *Tejero-Cantero, Alvaro, et al. "SBI--A toolkit for simulation-based inference." arXiv preprint arXiv:2007.09114 (2020).*
> **[7]** *Lueckmann, J., et al. "Likelihood-free inference with emulator networks. arxiv e-prints." arXiv preprint arXiv:1805.09294 (2019).*

---

> > ### Comment · Reviewer_GiCU · 2025-11-26
> > **Response**
> >
> > Dear Authors,
> >
> > thank you for the response. I am glad that the error in Appendix F2 can be attributed to a simple error, and I will revise my score to 4.
> >
> > Overall, I am still concerned regarding the conclusions drawn with respect to the batch MI rule. To me, it seems that it performs not significantly (and certainly not substantially) better than "just" ensembling (see also, e.g., table 5).

---

### Official Review · Reviewer_KpYD · 2025-11-03

**Soundness:** 2
**Presentation:** 2
**Contribution:** 3
**Rating:** 4
**Confidence:** 4

**Summary:**

In this study, the authors develop a sequential method for simulation-based inference (SBI) called "Ensemble sequential neural posterior estimation" (ESNPE). The method leverages ensembles of density estimators and in two variants of the method, a mutual information-based aquisition function (a non-batched and a batched version). On several benchmark tasks (Two moons, Gaussian mixture, Bernoulli GLM, SIR, Lotka-Volterra), the different variants of the method are shown to perform better than state-of-the-art SBI methods as assessed with C2ST and MMD.

**Strengths:**

### Originality

To my knowledge, this is one of the first methods in simulation-based inference that systematically uses posterior ensembles. Previous work is for the most part appropriately discussed and cited (see further comments in weaknesses).

### Quality

The study is technically sound, although I have a few points I would appreciate the authors to address (see weaknesses).


### Clarity

The manuscript is generally clearly written.


### Significance

The results in this study suggest that the developed approach, through its acquisition function, is a valuable contribution towards scalable simulation-based inference.

**Weaknesses:**

My main concerns lie in the quality of the experiments and clarity of the presentation:

- Tables 1 and 2 show results for single $x_o$ values. Why not show the averaged results in Appendix G? These are likely more robust estimates of the methods' performances, so I would strongly recommend the authors to show those results instead;

- Also, and related to the point above, although the methods proposed show systematic improvements in performance compared to previous methods, these improvements are often not substantial, especially taking the error bars into account. I urge the authors to replace the tables by plots, such that the reader can better assess the quality of the results;

- The experiments were run for one simulation budget. In order to have a better assessement of the performance of the developed methods, it would be crucial to run analysis over different total numbers of simulations;

- Regarding MI-ESNPE-batch, the reported runtime of the acquisition is negligible, even with very large batch sizes. This seems at odds with results in the literature, since most batch acquisition methods scale poorly with large batches (e.g., BatchBALD). I would urge the authors to check their results and/or provide an explanation for these results;

- It is well known that the atomic loss of SNPE-C suffers from posterior leakage. Could the authors comment on that issue in their methods?

- It is currently unclear whether the step of resampling and rejuvenation is really needed. It would be beneficial is the authors ran an experiment without that step;

- Appendix G3 shows that increasing the number of rounds leads to worse performance. This is an important limitation of the method and should be discussed in the main sections of the manuscript.

- Although there is relatively little work on ensembling in SBI, to my knowledge there are at least two previous papers that use ensembles: Hermans et al. 2022 (https://arxiv.org/abs/2110.06581), Deistler et al. 2022 (https://arxiv.org/abs/2210.04815). It would thus be important to discuss the relation to these studies.


Minor comments on clarity:

- It would be good to introduce "D" formally, in addition to its definition in algorithm 1;

- Figure 1 is never referenced in the text;

- "in terms of its likelihood $p(\phi|D)$", I assume it was meant "in terms of its posterior $p(\phi|D)$";

- "The shared parameter "suggestion" step". The word "suggestion" seems confusing and not necessary;

- In Appendix F2, the authors switched two references: "In Järvenpää et al. (2019), the authors recognize explicitly..." should be "In Krouglova et al. (2025), the authors recognize explicitly..." ; and "In Krouglova et al. (2025), the authors present several means" should be "In Järvenpää et al. (2019), the authors present several means"

**Questions:**

- Could the authors include the averaged results (as plots) over multiple observations (as in Appendix G) in the main text? These seem more representative than single observation results.

- The experiments were conducted with a single simulation budget. How does performance change across different total numbers of simulations?

- The reported negligible runtime of MI-ESNPE-batch for large batch sizes seems inconsistent with prior work (e.g., BatchBALD). Could the authors clarify how they assessed runtime and check their implementation of the batch aquisition method?

- What is the impact of the rejuvenation/resampling step? An ablation study would be beneficial in this respect.

- Appendix G3 shows that increasing the number of rounds can degrade performance. Could the authors discuss this limitation in the main sections of the manuscript?

---

> ### Author Response · Authors · 2025-11-23
> **Rebuttal (1/2)**
>
> Thank you for your time and effort providing detailed feedback on our work. Please find our responses to your questions and comments below.
>
> **Note**: Where applicable, we prefix sections with `W-<x>` or `Q-<y>`to reference itemized comments in Weaknesses and Questions, respectively, numbered by the order in which they were mentioned. Related items are grouped accordingly.
>
> ---
>
> **[W-1]** This is a fair request, and we agree it makes sense to switch the two tables. The original intent was to present results across specific posterior slices as highlighted in Figure 3 (which specifically visualizes Two Moons). This demonstrates nuance within individual settings rather than marginalizing over every instance. We nevertheless agree the averaged results from Table 5 present a more holistic accounting of performance and are more suitable as a primary table; we will switch the two in our updated manuscript (and add complementary plots; see below).
>
> **[W-2, W-3], [Q-1, Q-2]** Typical round-wise performance plots were produced during experimentation, but given the limited simulation budget (due to computational constraints), distinction across methods was somewhat less revealing. This additionally makes comparison with the non-sequential methods non-trivial, so we opted for simpler, final-round inference metrics. We nevertheless believe results from Tables 1 and 2 would be better presented as round-wise plots, especially under an expanded simulation budget, providing more runway for distinction across methods. Such plots, even at a fixed budget, would go some way to address concern [W-3], as one can interpret metrics reported across rounds as variable simulation budgets. These plots will be substituted in for the tables in our final manuscript. *(Please note we are still executing these larger runs and our manuscript will be updated upon completion. See below for additional details on new experiments to be completed before the end of the discussion period.)*
>
> **[W-4], [Q-3]** The fast runtimes of the batch acquisition scheme are within expectations, and not strictly at odds with the literature. This is a direct tradeoff of the approximate greedy scheme: because we *must* randomly sample parameters to build our batch in order to preserve convergence guarantees, we never bump into the combinatorially huge space of possible batches. This is additionally made clear by the runtime analysis provided in Appendix F3, where the complexity of the acquisition scheme has only a linear dependence on the batch size, $O(BMNK)$. Methods like BatchBALD attempt to compute optimal batches *until* hitting practical computational constraints, after which MCMC sampling is used (see Section 3.4 and Appendix C in the BatchBALD paper).
>
> **[W-5]** Posterior leakage with SNPE-C can often be effectively mitigated with renormalization under the prior (i.e., reweighting after sufficiently broad rejection sampling). This was used across every inference run; levels of remaining leakage are roughly equivalent across posteriors produced by SNPE-C and ESNPE (both use the same number of samples to approximate leakage factors). It is additionally worth noting that leakage-free methods like TSNPE lag behind in terms of performance on the reported tasks and simulation budgets, implying at the very least that risk of leakage may be a worthwhile practical tradeoff.
>
> **[W-6], [Q-4]** Resampling and rejuvenation has the effect of selecting for more diverse ensemble components. Without at least the  weighted likelihood bootstrap (WLB) sampling and variation in model initialization, one can expect each ensemble component to follow the same trajectory through model space, in which case we’d simply recover the vanilla SNPE-C performance. For large $K$, random weight initialization alone may be tenable, but at the ensemble sizes in our experiments (i.e., $K\le 16$), rejuvenation is effectively necessary to encourage diversity. This is alluded to in Appendix E, but we agree a more systematic analysis of its effects is warranted.
>
> **[W-7], [Q-5]** The behavior under increasing numbers of rounds reported in Appendix G3 is more indicative of dynamics at this specific simulation budget rather than a method limitation. As mentioned in the corresponding analysis, this is downstream of increasingly noisy round-wise updates, and an identical trend is observed for vanilla SNPE-C at this simulation budget. For larger budgets, one can expect to see more rounds lead to better metrics (up to a point), as the *absolute size* of the batch collected each round is important for model stability.

---

> ### Author Response · Authors · 2025-11-23
> **Rebuttal (2/2)**
>
> **[W-8]** Deistler et al. 2022 (i.e., TSNPE) is discussed at some length and is used as a core benchmark in our experiments. To the best of our knowledge, in the corresponding paper [3] the authors’ only use of ensembling occurs within the “multicompartment model of a single neuron” environment, in which case methodology defers the technique presented in Hermans et al. 2022. That is to say, ensembling is not a canonical component of the TSNPE method, and when used, is not distinct from the discussed approach Hermans et al. 2022.
>
> In Hermans et al. 2022 [5], ensembles are applied indiscriminately across several inference methods as a means of addressing posterior overconfidence. While there is indeed some overlap, this work does not prescribe a specific means of systematically maintaining an ensembled posterior across multiple rounds of inference. We nevertheless agree this work should be discussed in some capacity and appreciate the notice; our manuscript will be updated to incorporate this.
>
> **Minor comments**
> 1. Thanks for pointing this out. Some terms are used prior to full definitions in the introduction, but $D$ can be formally defined when setting up samples on line 83, i.e., $D = \{(\theta_i, x_i)\}_{i=1}^N$.
> 2. Thank you for mentioning this; Figure 1 should be referenced on line 210. This will be added.
> 3. This is indeed a posterior, but “likelihood” was used here in the generic sense, as in the quantity provided by a distribution at a point. Unfortunately the term is fairly overloaded in this context; we’ll revise the wording here.
> 4. Agreed, the wording in lines 250-251 is not particularly clear. This will be revised.
> 5. Thanks for calling attention to this; the citations on bullets 1 and 3 are indeed permuted. This discussion was originally in the main text but was moved to Appendix F2 due to space constraints and references were accidentally swapped in transition. This will be corrected.
>
> ---
>
> ### **Summary of new experimentation**
> *(this takes quite some time on our hardware; we appreciate your patience as we collect results to update our manuscript)*
>
> - We are running existing experiments at larger simulation budgets (~10x current max sample size) and are recording round-wise results in order to produce plots rather than purely tabular, final-round metrics.
> - We’re including two higher dimensional real-world environments to shed light on outstanding scalability concerns: the stomatogastric ganglion pyloric network [1] (31-dim $\theta \rightarrow$ 18-dim $x$) and the Hodgkin-Huxley model [2] (7-dim $\theta \rightarrow$ large noisy time series). Although these environments lack reference posterior samples for clean posterior quality evaluations (such as C2ST), they are widely analyzed across SBI literature (e.g., in [3], [4]) and a qualitative precedent exists when evaluating posterior predictive coverage.
>
> We’re more than happy to address any remaining concerns or provide additional clarification where needed.
>
> ---
>
> **References**
> **[1]** *Deistler, Michael, Jakob H. Macke, and Pedro J. Gonçalves. "Energy-efficient network activity from disparate circuit parameters." Proceedings of the National Academy of Sciences 119.44 (2022): e2207632119.*
> **[2]** *Gonçalves, Pedro J., et al. "Training deep neural density estimators to identify mechanistic models of neural dynamics." elife 9 (2020): e56261.*
> **[3]** *Deistler, Michael, Pedro J. Goncalves, and Jakob H. Macke. "Truncated proposals for scalable and hassle-free simulation-based inference." Advances in neural information processing systems 35 (2022): 23135-23149.*
> **[4]** *Gloeckler, Manuel, et al. "All-in-one simulation-based inference." arXiv preprint arXiv:2404.09636 (2024).*
> **[5]** *Hermans, Joeri, et al. "A trust crisis in simulation-based inference? your posterior approximations can be unfaithful." arXiv preprint arXiv:2110.06581 (2021).*

---

### Meta-Review · Area_Chair_XoCT · 2026-01-05

**Summary:**

The reviewers identify several significant concerns. Reviewer KpYD (score: 4) notes that while the method is technically sound and novel, experimental improvements over baselines are marginal and often within error bars. Reviewer GiCU (score: 4, revised from 2) initially raised concerns about citation errors and poor baseline performance but revised their score after the citation issue was clarified as a simple error. However, they maintain that the method "performs not significantly (and certainly not substantially) better than 'just' ensembling." Reviewer dTBE (score: 4) criticizes the use of simple benchmarks that don't reflect challenging SBI tasks and notes selective literature citation. Reviewer p85r (score: 4) questions the validity of the residual MI approximation and observes that active variants sometimes underperform non-active ones. All reviewers maintain scores at 4 (marginally below acceptance threshold).

**Reviewer Concerns:**

The authors address some concerns but leave critical issues unresolved. The citation error in Appendix F2 was satisfactorily explained. The authors commit to additional experiments with larger budgets and higher-dimensional tasks, which would partially address concerns about benchmark simplicity. However, the fundamental question raised by multiple reviewers remains: do performance gains come primarily from ensembling rather than MI-based acquisition? The authors acknowledge that "the ensembling and batch selection work in tandem" but provide no clear ablation, and Table 1 shows ESNPE (non-active) often performs comparably to MI variants. The authors' explanation that "there is noise across these strategies" concedes rather than resolves this limitation. Concerns about runtime comparisons and the validity of the residual MI approximation receive only partial responses. Critically, the authors request patience for new experiments to "update our manuscript," but these cannot be evaluated in the current process.

**Reviewer Scores:**

Reviewer KpYD would maintain their score of 4, as promised additional experiments don't address that "improvements are often not substantial, especially taking the error bars into account."

Reviewer GiCU revised from 2 to 4 after the citation clarification but remains "concerned regarding the conclusions drawn with respect to the batch MI rule," suggesting they would maintain 4.

Reviewer dTBE states they will "keep my score for now," suggesting 4 would persist.

Reviewer p85r's concerns about validity and inconsistent performance remain unaddressed, likely maintaining 4.

Average hypothetical score: 4. Recommendation: reject.

---

### Decision · Program_Chairs · 2026-01-26

Reject